# Eighty-eight variants highlight the role of T cell regulation and airway remodeling in asthma pathogenesis

Thorunn A. Olafsdottir[1,2], Fannar Theodors[1], Kristbjorg Bjarnadottir[1], Unnur Steina Bjornsdottir[3,4], Arna B. Agustsdottir[1], Olafur A. Stefansson[1], Erna V. Ivarsdottir [1,5], Jon K. Sigurdsson[1], Stefania Benonisdottir[1], Gudmundur I. Eyjolfsson[6], David Gislason[4,7], Thorarinn Gislason[2,8], Steinunn Guðmundsdóttir[1], Arnaldur Gylfason[1], Bjarni V. Halldorsson [1,9], Gisli H. Halldorsson [1], Thorhildur Juliusdottir[1], Anna M. Kristinsdottir[1], Dora Ludviksdottir[2,7], Bjorn R. Ludviksson[2,10], Gisli Masson[1], Kristjan Norland[1], Pall T. Onundarson[2,11], Isleifur Olafsson[12], Olof Sigurdardottir[2,13], Lilja Stefansdottir[1], Gardar Sveinbjornsson[1], Vinicius Tragante [1,14], Daniel F. Gudbjartsson [1,5], Gudmar Thorleifsson[1], Patrick Sulem [1], Unnur Thorsteinsdottir[1,2], Gudmundur L. Norddahl[1], Ingileif Jonsdottir [1,2]* & Kari Stefansson [1,2]*

Asthma is one of the most common chronic diseases affecting both children and adults. We report a genome-wide association meta-analysis of 69,189 cases and 702,199 controls from Iceland and UK biobank. We find 88 asthma risk variants at 56 loci, 19 previously unreported, and evaluate their effect on other asthma and allergic phenotypes. Of special interest are two low frequency variants associated with protection against asthma; a missense variant in *TNFRSF8* and 3' UTR variant in *TGFBR1*. Functional studies show that the *TNFRSF8* variant reduces TNFRSF8 expression both on cell surface and in soluble form, acting as loss of function. eQTL analysis suggests that the *TGFBR1* variant acts through gain of function and together with an intronic variant in a downstream gene, *SMAD3*, points to defective TGFβR1 signaling as one of the biological perturbations increasing asthma risk. Our results increase the number of asthma variants and implicate genes with known role in T cell regulation, inflammation and airway remodeling in asthma pathogenesis.

[1] deCODE genetics/Amgen, Inc., Reykjavik, Iceland. [2] Faculty of Medicine, School of Health Sciences, University of Iceland, Reykjavik, Iceland. [3] Department of Medicine, Landspitali, The National University Hospital of Iceland, Reykjavik, Iceland. [4] The Medical Center Mjodd, Reykjavik, Iceland. [5] School of Engineering and Natural Sciences, University of Iceland, Reykjavik, Iceland. [6] The Laboratory in Mjodd, RAM, Reykjavik, Iceland. [7] Department of Respiratory Medicine and Sleep, Landspitali, The National University Hospital of Iceland, Reykjavik, Iceland. [8] Department of Sleep, Landspitali, The National University Hospital of Iceland, Reykjavik, Iceland. [9] School of Science and Engineering, Reykjavik University, Reykjavík, Iceland. [10] Department of Immunology, Landspitali, The National University Hospital of Iceland, Reykjavik, Iceland. [11] Department of Laboratory Hematology, Landspitali, The National University Hospital of Iceland, Reykjavik, Iceland. [12] Department of Clinical Biochemistry, Landspitali, The National University Hospital of Iceland, Reykjavik, Iceland. [13] Department of Clinical Biochemistry, Akureyri Hospital, Akureyri, Iceland. [14] Department of Cardiology, Division Heart & Lungs, University Medical Center Utrecht, University of Utrecht, Utrecht, The Netherlands. *email: ingileif.jonsdottir@decode.is; kari.stefansson@decode.is

A sthma is one of the most common chronic diseases and has a substantial impact on the quality of life of both children and adults. Currently, it is estimated that around 300 million people have asthma worldwide and those numbers are predicted to rise in the coming years[1]. Asthma is a syndrome with heterogeneous pathophysiology and different asthma phenotypes differ in age of onset, environmental risk factors, clinical presentation, prognosis, and response to therapies[2]. To further characterize the distinct mechanistic pathways underlying the disease, asthma endotypes have been defined and can broadly be regarded as type 2 (T2) high or T2 low. T2 high asthma is characterized by increased activation of T helper cells of type 2 (Th2), innate lymphoid cells of type 2 (ILC2), and eosinophils. In contrast to T2 high, T2 low endotype is less well defined and is typically characterized by the absence of markers of T2 high disease and has rather been linked with activation of neutrophils, Th1 and/or Th17 cells[3]. This heterogeneity may explain why fewer asthma loci have been identified through genome-wide association studies (GWAS) than in other diseases of similar prevalence[4]. In total, 44 sequence variants (36 in Europeans and 8 in other ancestries) have been reported to associate with asthma in 27 independent GWAS[5-7]. We have previously discovered common sequence variants in *IL33* and it's receptor *IL1RL1* conferring risk of asthma[8], followed by an identification of a rare loss of function variant in *IL33* that protects against asthma[9], thereby supporting its relevance as pharmacological target for asthma.

Comorbidity between asthma and other allergic diseases (especially allergic rhinitis and atopic dermatitis) has been reported[10] and recent publications have focused on shared risk variants and genetic links between these traits[6,11].

Here we describe a large meta-analysis of asthma and report 88 independent associations at 56 loci. We perform a series of functional analysis to explore the biological effect of a low frequency missense variant in a gene of the tumor necrosis receptor family, *TNFRSF8*. We also report a low frequency 3 prime untranslated region (UTR) variant in *TGFBR1* that changes a microRNA (miR) recognition site and associates with increased *TGFBR1* expression in blood. Further, we report evidence of a single candidate gene for 8 of the 19 previously unreported asthma variants by extensive study of coding variants, expression quantitative trait loci (eQTLs) as well as enhancer and promoter signals. Lastly, we investigate association of the asthma variants with asthma sub-phenotypes (early-/late-onset and allergic asthma) as well as related traits (eosinophil count and allergic diseases).

## Results and discussion

### Genome-wide meta-analysis.
We performed a meta-analysis combining asthma GWAS results from Iceland ($n = 16,247$ cases, $n = 346,486$ controls) and the UK biobank ($n = 52,942$ cases, $n = 355,713$ controls). The asthma phenotype in both sets was based on physician diagnoses and/or self-reported doctor's diagnosis of asthma. Association between genotype and phenotype was tested by logistic regression, assuming a multiplicative model (see methods). We used genome-wide significant thresholds dependent on variant annotation and found association with asthma at 55 loci (>1 Mb apart) in addition to the extended HLA region (chr6: 25,000,000–35,000,000, build hg38). Conditional analysis revealed one or more secondary signals ($P < 5 \times 10^{-8}$) at 17 of the 56 loci thus yielding 88 independent signals (Fig. 1, Supplementary Data 1). Based on number of cases and controls, we note that the effective sample size is four times bigger for the UK biobank samples ($N = 84,396$) than the Icelandic ($N = 22,689$), hence most of the power comes from the UK biobank samples.

The estimated genomic inflation in the two studes (estimated using LD-score regression)[12] is 1.368 for the Icelandic dataset and 1.092 for the UK biobank dataset, reflecting the relatedness of individuals in Iceland which is also the most likely explanation of genomic inflation in UK biobank where 30% of the participants have a relative (third degree or closer) in the UK biobank dataset[13].

We note that only 4 out of the 88 variants showed significant heterogeneity in effect sizes based on the number of variants tested ($P < 0.05/8 = 5.62 \times 10^{-4}$) and 28 showed nominal heterogeneity ($P < 0.05$) between the two sample sets demonstrating a good consistency of effect for vast majority of the variants. The observed heterogeneity of the effect estimates is likely due to difference in the phenotype definition and/or ascertainment between the studies, coherent with retrospective studies as reported here. Further, 85 out of the 88 variants ($P < 1 \times 10^{-16}$) had effect size in the same direction in the two cohorts and 44 were at least nominally significant in Iceland showing good replication of effects (Supplementary Data 1).

Of the 88 independent signals, 47 are at 24 previously reported asthma loci, 22 variants at 16 loci have previously been reported for a combined allergic phenotype[11] and 19 variants at 16 loci are previously unreported asthma signals. We note that while this manuscript was under consideration two independent reports published association of asthma with 9 of those 19 variants as indicated in Table 1[14,15].

Out of the 47 variants at previously reported loci, 24 were represented by previously reported variants ($r^2 \geq 0.2$; Supplementary Data 2) and 23 by previously unreported signals at those loci ($r^2 < 0.2$; Supplementary Data 3). Furthermore, we replicated 31 out of the 36 reported European asthma loci[7,16] when adjusting for the 36 variants tested ($P < 0.05/36 = 1.4 \times 10^{-3}$). For two of five non-replicating variants we found other genome wide significant (GWS) variants at the respective loci (Supplementary Data 4).

Forty-one of our 88 variants at 32 loci have not been previously associated with asthma, although 22 variants at 16 loci have previously been reported for a combined allergic phenotype of asthma, hay fever and eczema[11] (Supplementary Data 5).

Altogether, we identified 19 independent signals at 16 loci not previously reported to associate with asthma or the combined allergic phenotype[11] (Table 1). Sixteen of the previously unreported variants were common (Effect Allele Frequency (EAF) ≥ 5%) and three were low frequency variants. None of those variants showed significant ($P < 0.05/88 = 5.62 \times 10^{-4}$) heterogeneity of effects in the two sample sets (Table 1).

**Loss of function variant in *TNFRSF8*.** One of the low frequency variants (EAF = 1.2% (ICE)/1.5% (UK)) rs2230624_A is a missense variant p.Cys273Tyr in *TNFRSF8 (alias CD30)* that associates with reduced asthma risk. The p.Cys273Tyr variant had the greatest protective effects (OR = 0.82, $P = 8.27 \times 10^{-13}$) of the 19 previously unreported variants and in fact there were only 4 out of the 88 asthma variants with greater effect on asthma risk; rs72782676 (intergenic on chr10, OR = 0.63), rs149045797 (intron variant at *IL33* locus, OR = 0.65), rs12722502 (intron variant at *IL2RA* locus, OR = 0.80 and rs61816761 (stop-gained variant at *FLG* locus, OR = 1.24). The rs2230624_A variant is a singleton (all LD < 0.4; Fig. 2a) and is predicted by PROSITE database[17] to disrupt a disulfide bond between Cys in positions 273 and 259 in the extracellular domain of the protein. CD30 is expressed on the surface of activated lymphocytes and eosinophils and has been implicated in activation, proliferation and apoptosis via NFκB activation[18-21]. p.Cys273Tyr has been reported to associate with reduced eosinophil count[22] and reduced mosquito

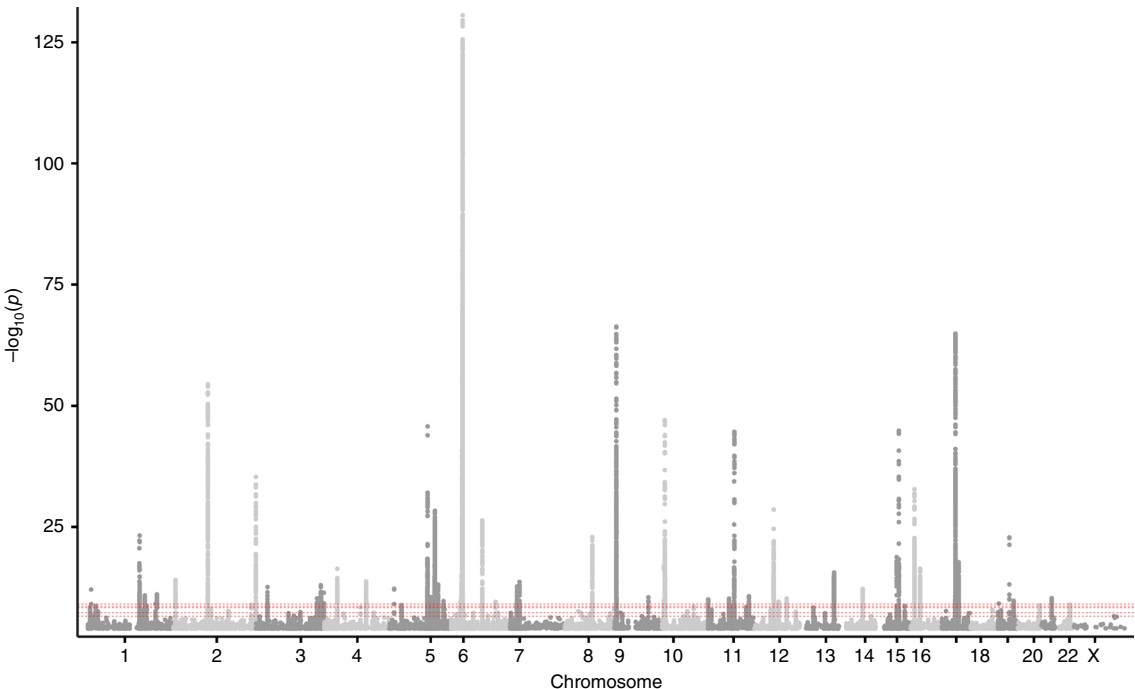

**Fig. 1 Sequence variants associating with asthma.** Manhattan plot for the Iceland-UK BB meta analyses of asthma ($N_{cases} = 69,189$). In all, 56 regions harbor genome-wide significant signals. Variants are plotted by chromosomal position (x-axis) and $-\log_{10}P$ values (y-axis). Dotted line indicate the different $P$ value thresholds applied based on variant annotation. The adjusted significance thresholds are represented by horizontal dashed line from bottom to the top in the following order: $2.6 \times 10^{-7}$ for variants with high impact ($N = 8,464$), $5.1 \times 10^{-8}$ for variants with moderate impact ($N = 149,983$), $4.6 \times 10^{-9}$ for low-impact variants ($N = 2,283,889$), $2.3 \times 10^{-9}$ for other variants in DNase I hypersensitivity sites ($N = 3,913,058$) and $7.9 \times 10^{-10}$ for all other variants ($N = 26,108,038$).

bite size[23]. Increased soluble CD30 (sCD30) in serum has been associated with increased severity of asthma in children[24] and CD30 knock-out mice are protected against asthma[25]. Therefore, we postulate that the missense p.Cys273Tyr in *TNFRSF8* that associates with decreased asthma risk reduces the function of CD30. In order to investigate this, we over-expressed wild-type (WT) CD30 or the p.Cys273Tyr variant in HeLa cells and compared levels of CD30 protein generated. CD30 is expressed as a precursor protein that undergoes post-translational modification, that turns it into the mature form of the protein[26,27]. Lysates from cells expressing the p.Cys273Tyr variant had higher ratio of the precursor form to the mature form than that observed in cells expressing the WT CD30 ($P = 4.2 \times 10^{-7}$, two-tailed paired *t*-test; Fig. 2b and Supplementary Fig. 1). Moreover we observed both a lower cellular surface expression on p.Cys273Tyr than WT CD30 cells ($P = 7.4 \times 10^{-5}$, two-tailed paired *t*-test, Fig. 2c) and a significantly lower amount of sCD30 in the culture supernatants of cells expressing the p.Cys273Tyr variant ($P = 5.5 \times 10^{-4}$, two-tailed paired *t*-test, Fig. 2d). In line with this, cell surface expression of CD30 was significantly lower on in vitro stimulated peripheral blood mononuclear cells (PBMCs) derived from p. Cys273Tyr heterozygote carriers ($P = 1.5 \times 10^{-2}$, two-tailed Wilcoxon matched-pairs signed rank test; Fig. 2e). We recruited six p.Cys273Tyr homozygotes and matched non-carriers and measured CD30 surface expression both on stimulated PBMCs and Epstein-Barr virus transformed lymphoblasts. Surface expression of CD30 tended to be lower on EBV transformed lymphoblasts from homozygous carriers than from matched non-carriers ($P = 0.063$, two-tailed Wilcoxon matched-pairs signed rank test), whereas no difference was detected when comparing stimulated PBMCs from homozygous carriers to matched non-carriers ($P = 0.22$, two-tailed Wilcoxon matched-pairs signed rank test) (Supplementary Fig. 2). However, these results warrant

further investigation with larger number of homozygotes especially because of the inherent variance of the PBMC stimulation assay. Together our data suggest that the missense variant, leading to the disruption of a disulfide bond between Cys in positions 273 and 259, reduces trafficking of the protein to the cell surface. Furthermore, reduced levels of sCD30 were observed in the cell culture supernatant of in vitro stimulated PBMCs from heterozygote carriers ($P = 3.1 \times 10^{-5}$, two-tailed Wilcoxon matched-pairs signed rank test) compared with those from non-carriers (Fig. 2f). This is consistent with reduced surface-expression although an effect of the variant on the CD30 shedding itself cannot be excluded[28].

**Gain of function variant in *TGFBR1*.** Another low frequency variant of special interest, is a 3 prime UTR variant, rs41283642_T (EAF = 1.9% (ICE)/3.4% (UK)), in *TGFBR1* that associated with reduced asthma risk ($P = 2.16 \times 10^{-10}$, OR = 0.89). eQTL analysis showed that in blood the variant associated with 15.2% increase in *TGFBR1* expression per allele ($P = 7.24 \times 10^{-101}$, effect = 1.18 SD, which is the most significant eQTL for *TGFBR1*; Fig. 3a, b, Supplementary Data 6). rs41283642_T changes a recognition site for microRNA (miR) miR-142-3p (Fig. 3c). miR-142-3p has previously been shown by others[29] to bind to the 3 prime UTR site in TGFβR1 and therby repress TGFβR1 expression at both RNA and protein levels. Interestingly, miR-142-3p is one of only three miRs reported to have an increased expression in severe asthmatic lungs[30]. Together these data suggest that rs41283642_T reduces the binding of miR-142-3p to TGFβR1 transcripts leading to increased TGFβR1 expression and through that protection against asthma.

**Unreported common asthma risk variants.** Four of the 16 previously unreported common variants are missense, rs2228552_T in

**Table 1 Previously unreported loci associated with asthma in Iceland – UK biobank meta analyses.**

| Lead SNP | chr:Position | Allele (EA/OA) | EAF % (ICE/UK BB) | Coding effect | Gene | Coding change | OR (95% CI) | P | P_het |
|---|---|---|---|---|---|---|---|---|---|
| rs2230624[a] | chr1:12115601 | A/G | 1.2/1.5 | Missense | TNFRSF8 | Cys273Tyr/Cys162Tyr | 0.82 (0.78–0.87) | 8.3E-13 | 0.14 |
| rs2228552[a] | chr1:31699894 | T/G | 68.3/64.3 | Missense | COL16A1 | Thr62Lys/Thr52Met | 1.04 (1.02–1.05) | 2.6E-08 | 0.86 |
| rs2296618 | chr1:198697103 | G/A | 14.4/13.5 | Downstream | PTPRC[a] | – | 0.94 (0.93–0.96) | 6.9E-10 | 0.36 |
| rs7626218[a] | chr3:177134250 | T/A | 41.1/39.5 | Intron | TBL1XR1[a] | – | 0.96 (0.95–0.97) | 5.9E-11 | 0.14 |
| rs34712979 | chr4:105897896 | A/G | 24.4/25.8 | Splice region | NPNT | – | 1.04 (1.03–1.06) | 3.9E-09 | 0.85 |
| rs11746314[a] | chr5:157325949 | G/A | 3.6/5.8 | Intron | CYFIP2[a] | – | 1.09 (1.06–1.12) | 2.1E-10 | 0.832 |
| rs3813308 | chr5:119355086 | G/C | 47.3/43.5 | 5 prime UTR | TNFAIP8 | – | 1.04 (1.03–1.06) | 2.83E-11 | 0.8 |
| rs1800797 | chr7:22726602 | G/A | 52.2/58.0 | Upstream | IL6[a] | – | 1.04 (1.03–1.05) | 2.9E-10 | 0.66 |
| rs34173062 | chr8:144103704 | A/G | 6.6/7.2 | Missense | SHARPIN | Ser17Phe | 1.08 (1.05–1.10) | 1.0E-08 | 0.28 |
| rs41283642[a] | chr9:99153605 | T/C | 1.9/3.4 | 3 prime UTR | TGFBR1 | – | 0.89 (0.86–0.83) | 2.16E-10 | 0.05 |
| rs12788104[b] | chr11:1129831 | G/A | 73.4/68.7 | Intergenic | MUC2[c] | – | 1.04 (1.03–1.06) | 1.7E-10 | 0.09 |
| rs174562[a] | chr11:61817672 | G/A | 38.8/34.6 | Upstream | FADS1[c] | – | 0.96 (0.95–0.97) | 6.3E-11 | 0.31 |
| rs7961712[a] | chr12:9421187 | A/G | 86.3/85.0 | Intron | PLXNC1[c] | – | 1.06 (1.04–1.08) | 5.8E-11 | 0.45 |
| rs34939984 | chr16:27234391 | T/C | 40.1/35.9 | Intron | NSMCE1[c] | – | 1.04 (1.03–1.06) | 5.2E-11 | 0.41 |
| rs3024664 | chr16:27360103 | C/T | 93.5/93.9 | Intron | IL4R[c] | – | 1.12 (1.09–1.15) | 4.4E-17 | 0.23 |
| rs6498021 | chr16:27403057 | G/T | 85.3/86.0 | Intron | IL21R | – | 0.95 (0.93–0.97) | 2.8E-08 | 0.30 |
| rs179771 | chr16:27406423 | C/G | 53.9/48.8 | Intron | IL21R[c] | – | 1.04 (1.03–1.05) | 3.23E-09 | 0.08 |
| rs117552144 | chr19:3136093 | T/C | 5.3/6.7 | 5 prime UTR | GNA15 | – | 1.09 (1.06–1.11) | 6.7E-10 | 0.74 |
| rs8103278[a] | chr19:45867123 | A/G | 31.3/35.3 | Upstream | SYMPK[c] | – | 0.96 (0.95–0.97) | 1.8E-10 | 0.84 |

Results are shown for the combined sample-sets of Iceland and the UK BB. $P_{het}$ represents the statistical heterogeneity between the two sample-sets. Gene indicates the most likely candidate gene at the locus as described in Supplementary Data 7
EA effect allele, OA other allele, EAF effect allele frequency ICE Iceland, UK BB UK biobank, OR odds ratio
[a] loci reported in Johansson et al[15], while this paper was in review
[b] loci reported in Shrine et al[14], while this paper was in review
[c]The closest gene is indicated for loci where our analysis do not pinpoint the most likely gene candidate

COL16A1 (Thr62Lys/Thr52Met) and rs34173062_A in SHARPIN (Ser17Phe) (Table 1) or highly correlated ($r^2 > 0.8$) with coding variants in NSMCE1 (rs34939984) or SYMPK, SIX5 and BHMG1 (rs8103278) (Supplementary Data 7). COL16A1 encodes the α-chain of type XVI collagen, involved in integrity of the extracellular matrix in association with fibril forming collagens type I and II[31]. Increased local collagen type I production has been reported in asthma patients, possibly mediated by TGFβ secreting eosinophils[32]. The rs2228552_T associated with increased asthma risk ($P = 2.6 \times 10^{-8}$, OR = 1.04) and was highly correlated ($r^2 > 0.8$) with the top eQTL for increased COL16A1 expression in fibroblasts (Supplementary Data 8). Thus the COL16A1 association with asthma may be through its role in airway remodeling. SHARPIN encodes a component of the LUBAC complex, which plays a role in NFκB activation and regulation of inflammation[33]. The SHARPIN missense variant, Ser17Phe, associated with increased risk of asthma ($P = 1.00 \times 10^{-8}$, OR = 1.08) and increased eosinophil count in blood[22]. Furthermore, homozygous loss of function mutations in Sharpin mice induce extensive eosinophilic inflammation in multiple organs, including lung, esophagus and skin[34,35].

We found four independent signals at the IL4 receptor alpha (IL4Rα)/IL21R locus, including the intronic variant rs6498021_G in IL21R that was highly correlated ($r^2 > 0.8$) with the top eQTL signal for decreased IL21R expression in blood (Supplementary Data 6). Polymorphisms at the IL4Rα locus have been associated with asthma in candidate gene studies but not in GWAS[36–39]. Biologicals targeting IL4Rα show promise in the treatment of persistent asthma[40] but the role of IL21R in asthma has been less studied. Other loci of interest include NPNT, TNFAIP8, FADS1, GNA15, and IL6. Taken together, based on analysis of coding variants, eQTLs in different tissues, enhancer and promoter signals in CD4+ Th cells we found evidence of a single candidate gene for 8 out of the 19 previously unreported asthma variants (Supplementary Data 9).

**Signals at known asthma loci.** Among the 24 previously identified loci were IL1RL1/IL18R1, GATA3, TLR1, TSLP, HLA, IL4/IL5/IL13, RORA, GSDMB, SMAD3, and IL33[9,41]. Our top signal at the SMAD3 locus, rs72743461_A, correlated with reported asthma signals[16,42,43] (Supplementary Data 2) but we further identified 3 secondary signals (Supplementary Data 3), including rs117683492_A, that associated with increased asthma risk and was the top eQTL reducing SMAD3 expression in blood (Supplementary Tables 1 and 6). SMAD3 is a signaling molecule downstream of TGFβR and Smad3 KO mice show defective TGFβ-mediated repression of cytokine production and cell proliferation[44]. Together, the TGFBR1 3 prime UTR variant rs41283642_T described above and rs117683492_A at the SMAD3 locus point to increased risk of asthma in individuals with defective TGFβR1 signaling in line with its known role in immunosuppression[45]. rs6926894 is the top variant in the HLA region ($P = 2.5 \times 10^{-126}$, OR = 1.16). Given the complicated LD in the HLA region we tested association of HLA alleles with asthma in the Icelandic dataset finding strongest association with HLA-DRB1*04 ($P = 2.06 \times 10^{-25}$) that largely explains the rs6926894 association (Supplementary Data 10).

**Association of asthma variants with eosinophil count.** We have previously used the well-established link between eosinophilic airway inflammation and asthma to discover common sequence variants in IL1RL1 and IL33 that associate strongly with blood eosinophil counts and risk of asthma[8]. Therefore, we tested the association of our 88 asthma variants with eosinophil blood counts in Iceland and UK biobank. Of those, 69 associate significantly with eosinophil count after adjusting for the number of

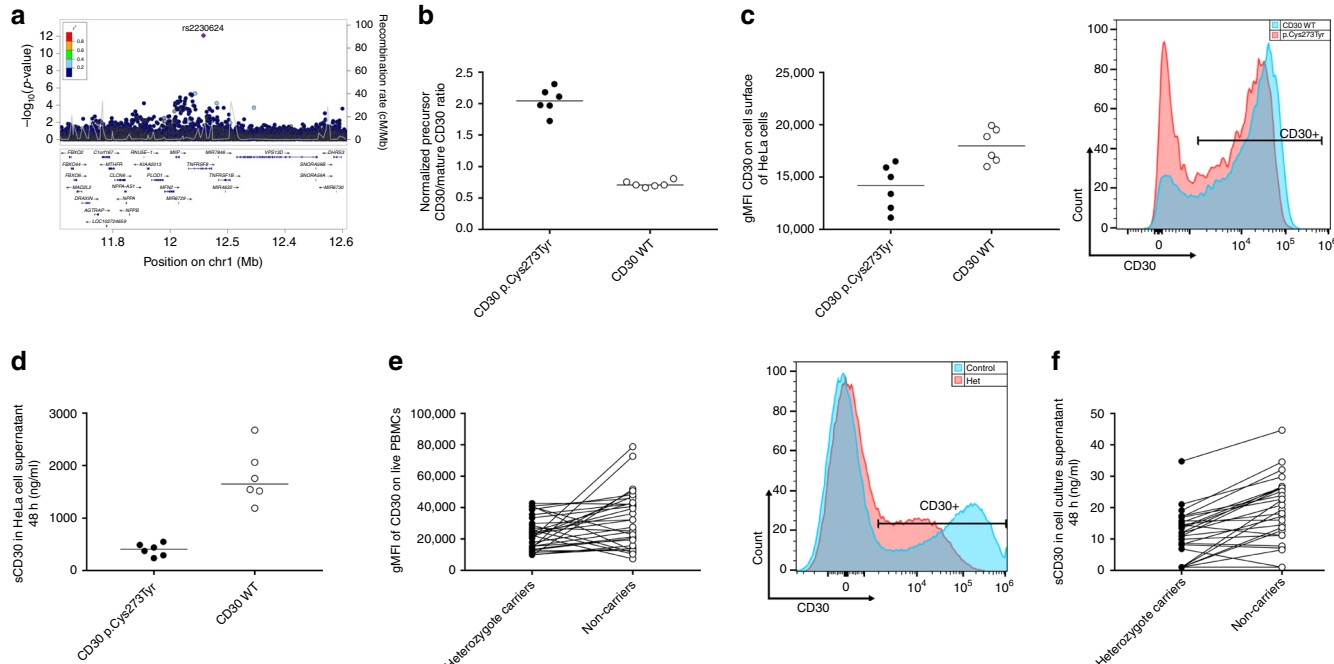

**Fig. 2 Variant associating with reduced asthma risk affects protein expression and shedding of CD30. a** Locus plots showing association with asthma where the lead variant p.Cys273Tyr (rs2230624) is colored in purple. Other variants are colored by degree of correlation ($r^2$) with the lead variant. **b** Protein simple WES analysis of CD30 expression in cell lysates from HeLa cells overexpressing CD30 wild-type or variant p.Cys273Tyr CD30; graph showing ratio of CD30 precursor/mature intensity. **c** Surface expression of CD30 on HeLa cells over-expressing wild-type or p.Cys273Tyr CD30 measured by flow cytometry, displayed as geometric mean of fluorescense intensity (gMFI). Histogram showing a representative CD30 surface expression p. Cys273Tyr (red) and CD30 WT (blue). **d** sCD30 levels (ng/ml) in cell culture supernatant from HeLa cells over-expressing wild-type or p.Cys273Tyr CD30. **e** Surface expression of CD30 on stimulated PBMCs from heterozygous p.Cys273Tyr carriers and age and gender matched non-carriers, measured by flow cytometry, displayed as gMFI. Histogram showing a representative CD30 surface expression from one pair of PBMCs in heterozyogus CD30 p.Cys273Tyr carrier (red) and age and gender matched non-carrier (blue). **f** sCD30 levels (ng/ml) in cell culture supernatant of PBMCs from p.Cys273Tyr heterozygotes and non-carriers. The dots in plots **b**–**d** represent individual experiments. The dots in plots **e** and **f** represent individual donors. Lines in panel **b**–**d** indicate median level. In plots **e** and **f** the line between carriers and non-carriers indicated age and gender matched pairs.. Two-tailed Wilcoxon matched-pairs signed rank test was used to test for significant differences in PBMCs. Two-tailed paired *t*-test was used to test for significant differences in HeLa cells. Source data are provided as a Source Data file.

variants tested, with the asthma risk allele associating with higher eosinophil count for 67 of the variants ($P < 0.05/93 = 5.4 \times 10^{-4}$; Supplementary Data 11). Among the 20 variants that did not associate with eosinophil count and deviate from the overall correlation between asthma risk and eosinophil count are variants at the filaggrin family member *FLG/HRNR* locus, suggesting their role in non-eosinophilic asthma (Supplementary Fig. 3).

**Effect of asthma variants on different asthma sub-phenotypes.** Given the known heterogeneity of asthma we tested the effect of the asthma variants on three asthma sub-phenotypes, allergic asthma (AA), early (EOA) and late (LOA) onset asthma. Overall, the effects of the 88 asthma variants were similar on the sub-phenotypes although for some of the variants (e.g at *FLG/HRNR* locus) the effect was stronger on EOA than LOA (Fig. 4a). Further, the effect of *IL33* (rs149045797) variant was largest on EOA (OR = 0.53, $P = 3.8 \times 10^{-9}$) followed by LOA (OR = 0.71, $P = 5.1 \times 10^{-9}$) with smallest effect on AA (OR = 0.93, $P = 0.70$; Fig. 4 and supplementary Data 12). This variant is fully correlated ($r^2 = 1$) with a loss of function variant (rs146597587) previously reported to associate with reduced IL33 mRNA expression and asthma risk and in line with the results reported here, the largest effect was observed in young children who were hospitalized at least 4 times before the age of 6 years due to asthma exacerbation (OR = 0.24, $P = 0.04$)[9]. However, 58% of those young children had atopic diagnosis before 6 years of age[46] indicating that the IL33 association is driven by severe asthma rather than non-AA

exclusively. We note that we lack information on AA for a large part of our samples, especially in the UK biobank samples where only 542 (1% of all asthma cases) have this diagnosis and there is an overlap both with the early and late onset asthma cases. Therefore, the IL33 association with asthma severity in AA and non-AA warrants further investigation.

Conversely, the effect of the *FLG* variant was stronger on AA than on LOA and the effect of a variant at the *GATA3* locus was stronger on AA than on either EOA or LOA (Fig. 4c). These results point to a partly distinct genetic architecture behind the three asthma phenotypes tested in our analysis in line with the clinical heterogeneity. Our results show that certain sequence variants have stronger effect in EOA than LOA indicating that genetics may play a larger role in EOA.

**Genetic correlation between asthma and allergic phenotypes.** Recently, 136 variants were reported to associate with a combined allergic phenotype (asthma, hay fever and/or eczema)[11]. The overall effects of these 136 allergy variants correlated well with the effects in our asthma meta-analysis, although several markers (e.g. *TSLP*, *GATA3*, *SMARCE1*) demonstrated weak correlation (Supplementary Fig. 4). This is in line with reported genetic correlation between asthma and allergic diseases (hay fever/ allergic rhinitis or eczema)[6].

We also tested the 88 independent asthma variants for association with different allergic phenotypes, (Supplementary Table 1). Interestingly, the effects of these variants on asthma

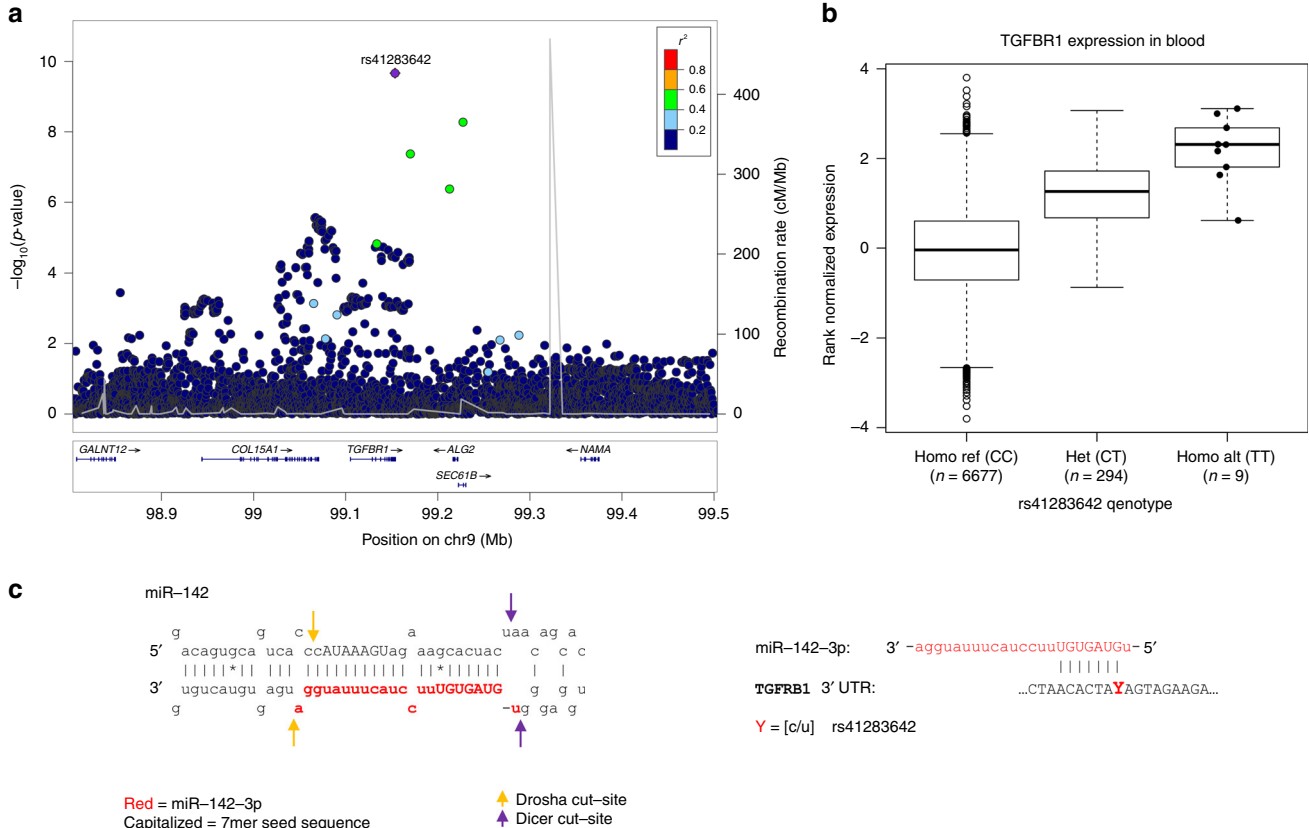

**Fig. 3 Variant in the 3'UTR of TGFBR1 increases TGFBR1 expression in blood and associates with reduced asthma risk. a** Locus plot showing association with asthma where the lead variant rs41283642 is colored in purple. Other variants are colored by degree of correlation ($r^2$) with the lead variant. **b** Box plot showing the expression of TGFβR1 in blood of rs41283642_T non-carriers (CC, $N = 6677$), heterozygotes (CT, $N = 294$) and homozygotes (TT, $N = 9$) based on RNA-Seq data; $P = 7.24E\text{-}101$, effect = 1.18 S.D. corresponding to 15.2% increased expression of TGFBR1 per allele. The bottom and top of the boxes correspond to the 25th (Q1) and 75th (Q3) percentiles, the line inside the box corresponds to the median, and the whiskers are located at max(min (Expression), Q1 – 1.5 IQR) and min(max(Expression), Q3 + 1.5 IQR), respectively (where IQR is the interquartile range = Q3 – Q1). **c** Schematic representation of the primary miR-142 transcript, derived from MIR142 located on chromosome 17q22. The 3'end of miR-142 forms miR-142-3p which is known to targets multiple mRNA transcripts including TGFRB1. The predicted binding site for miR-142-3p at the 3'UTR of TGFBR1 is shown wherein Y represents the location of the rs41283642. The reference allele is C, forming a 7mer perfect complementary sequence to the seed sequence of miR-142-3p whereas the protective asthma T_allele (U in RNA), disrupts the 7mer binding site. Source data are provided as a Source Data file.

correlated significantly with their effects on allergic rhinitis (AR; $P = 3.1 \times 10^{-13}$, $r^2 = 0.68$; Supplementary Fig. 5a) but not with atopic dermatitis (AD; $P = 9.80 \times 10^{-2}$, $r^2 = 0.18$; Supplementary Fig. 5b). Out of the 88 asthma loci, two variants at the same locus, a loss of function variant in *FLG* (rs61816761) and intergenic variant (rs12123821,close to *HRNR*) had greatest effect on AD (Supplementary Data 12). Further, significant correlation was observed between effects of the asthma variants on asthma and on nasal polyps (NP) as well as on chronic rhinosinusitis (CRS) with NP (CRSwNP) but not with CRS without NP (CRSsNP) in agreement with differences in the pathogenesis of these closely related phenotypes[47] (Supplementary Fig. 5c–f). Variants at *IL33*, *GATA3*, *BACH2*, *HLA* and *TSLP* loci significantly associated with >3 asthma/allergic phenotypes (($P < 0.05/(88 \times 9) = 5.7 \times 10^{-5}$), adjusted for number of variants and phenotypes tested; Supplementary Data 12), most of which are important for generation and function of ILC2 and Th2[48]. Taken together, the correlation of effects of the 88 asthma variants was stronger with respiratory allergic phenotypes of T2 than with AD or the Th1 driven CRSsNP. However, we observed substantial overall genetic correlation between asthma and AD as well as between asthma and allergic rhinitis indicating that other variants than the 88 reported here play a significant role in the genetic link between

asthma and AD (Supplementary Table 2). Since this is a retrospective study, not designed to study co-morbidities of asthma and allergic diseases, we cannot exclude that some of the overlapping association of certain sequence variants with e.g. asthma and AD might be due to overlap of the two phenotypes. However, only 980 out of 16,247 asthma cases in Iceland had the AD diagnosis (6%) while AD was hardly reported as a single disease in UK biobank, making it impossible to accurately define asthmatics without AD.

**Pathway analysis.** Pathway analysis was performed using DEPICT[49] to search for the biological connectivity between the asthma association signals. DEPICT prioritized 71 (FDR < 0.05) known genes (Supplementary Data 13) and identified 787 significantly (FRD < 0.01) enriched gene sets, the majority involved in T cell biology, mainly implicating CD4$^+$ T cells, regulation of their activation, responses and physiology (Supplementary Data 14).

In summary, our study considerably expands the number of asthma susceptibility loci and confirms many previous findings. The results highlight the role of Th cells in asthma in line with imbalanced T cell regulation reported to play a critical role in

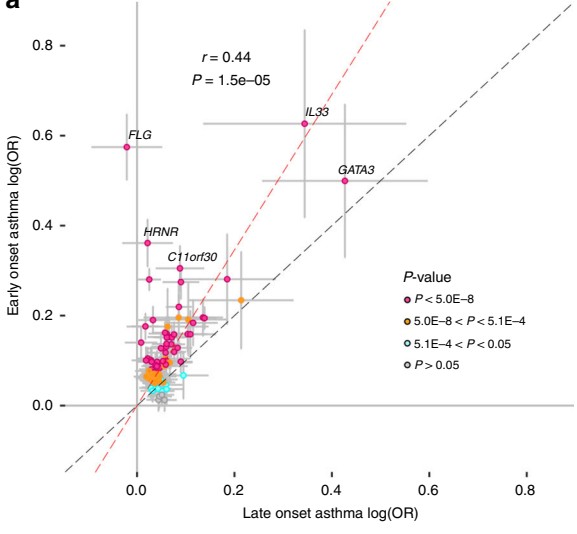

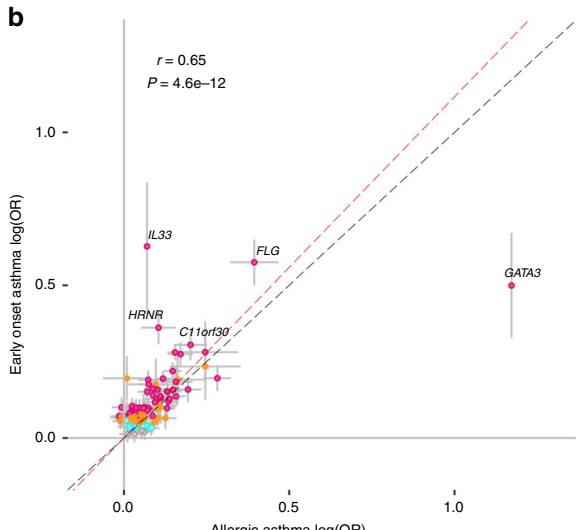

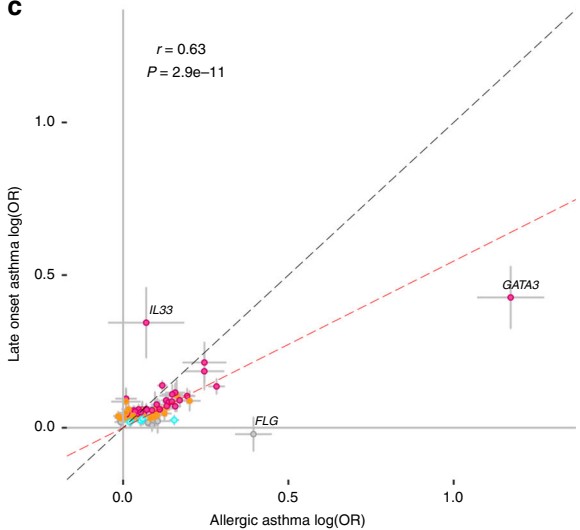

**Fig. 4 Effect of asthma associating variants in asthma sub-phenotypes.** Early onset asthma (EOA), Late Onset Asthma (LOA) and Allergic Asthma (AA). The x-axis and the y-axis show the logarithm of the estimated odds ratios for **a** EOA and LOA, **b** AA and EOA, and **c** AA vs LOA, respectively. All effects are shown for the asthma risk increasing allele based on meta-analysis of the Icelandic and the UK BB sample sets. Error bars represent 95% confidence intervals. The red line represents results from a simple linear regression through the origin using MAF (1-MAF) as weights and the gray line indicates the reference line with slope = 1. The weighted correlation coefficients (r) and P values (t-test) are shown in the graphs. Source data are provided as a Source Data file.

## Methods

**Study sample sets for asthma, allergy phenotypes, and eosinophil count.** Asthma in the UK biobank was defined as ICD10 diagnoses in fields 41202 or 41204, including anyone of J45.0, J45.1, J45.8, J45.9, and J46 and/or self-reported by the non-cancer illness code, self-reported during verbal interview (data-field 20002) with a code for asthma (1111).

Icelandic asthma patients over 18 years of age were recruited who attended an asthma clinic or emergency rooom at the National University Hospital of Iceland or the Icelandic Medical Center (Laeknasetrid) during the years 1977 to 2017. Asthma diagnosis was based on a combination of physician's diagnosis and ICD10 diagnosis, including anyone of J45.0, J45.1, J45.8, J45.9, and J46 and/or self-reported by a positive reply to the question: has a doctor confirmed your asthma diagnosis. Atopy status determined by skin prick testing and age of onset was available for part of the asthma cohort both in Iceland and UK biobank. Early onset was defined as first diagnosis ≤17 years of age and late onset as first diagnosis >17 years of age. Late onset asthma diagnosis from Iceland contains both where the first diagnosis was at or after 18 years of age and where the first diagnosis is unknown. In total we had 16,247 asthma cases and 346,486 controls from Iceland and 52,947 cases and 355,713 controls from the UK biobank. Estimating the effective number of cases as $2 \times Na \times Nc / (Na + Nc)/GC$, where Na and Nc are the number of cases and controls, respectively, and GC is the genomic adjustment factor, estimated using LD-score regression[12], we can calculate the effective sample size for Iceland to be $Neff = 22,689$ and UK biobank to be $Neff = 84,396$. $GC = 1.368$ in Iceland and $GC = 1.092$ in UK biobank. Characteristics of Icelandic and UK biobank asthma cohorts is further given in Supplementary Table 3.

Allergic rhinitis combines doctoral diagnosis of allergic rhinitis from Iceland and a questionnare data from UK biobank on hayfever or allergic rhinitis (Non-cancer-illness code 1387). As only 42 individuals had the ICD10 code for Atopic dermatitis (L20) in UK biobank we ran meta-analysis with our Icelandic list ($N = 8325$) derived both from physician's diagnosis and ICD10 code (L20) together with published meta-analysis on AD[52] downloaded from the GRASP database [https://grasp.nhlbi.nih.gov/FullResults.aspx], to study this phenotype. Other allergy diagnosis were based on ICD10 codes from UK biobank and either physician's diagnosis or ICD10 codes from Iceland. The allergy phenotypes used were: Nasal polyps (ICD10:J33), chronic sinusitis (ICD10:J32), chronic sinusitis with nasal polyps (combined: ICD10:J32 and J33), and chronic sinusitis without nasal polyps (ICD10:J32 without ICD10:J33). Number of genotyped individuals in each cohort are listed in Supplementary table 1. Icelandic controls were participants from various deCODE genetics programs.

We obtained eosinophil counts from three of the largest laboratories in Iceland (measurements performed between the years 1993 and 2015). The circulating eosinophil counts were standardized to a standard normal distribution using quantile-quantile standardization and then adjusted for sex, year of birth, and age at measurement, as previously described[8,9]. A total of 251,307 Icelanders with eosinophil counts were included in the study. In the UK biobank we had eosinophil counts for a total of 396,822 individuals that were adjusted for sex, year of birth, age at measurement, and the first 40 principal components, then inverse normal transformed.

All participating individuals who donated blood signed informed consent. The personal identities of participants were encrypted using a third-party system approved and monitored by the Icelandic Data Protection Authority[53]. The study was approved by the National Bioethics Committee in Iceland (Approval no. VSN 14-099).

**Whole-genome sequencing and imputation.** The GWASs in Iceland were performed with 32.5 million markers identified through whole-genome sequencing of 15,520 Icelanders to an average genome-wide coverage of 34X and subsequently imputed into 151,677 chip-typed individuals, as well as their first and second degree relatives. The imputation has been extensively described in recent publications[54]. Genotyping of UK biobank samples was performed using a custom-made Affymetrix chip, UK BiLEVE Axiom[55], and with the Affymetrix UK Biobank Axiom array[56]. Imputation was performed by the Wellcome Trust Centre for Human Genetics, using the Haplotype Reference Consortium (HRC) and the

asthma pathogenesis[50,51]. We show evidence that two low frequency variants in *TNFRSF8* and *TGFBR1* associate with decreased asthma risk through loss of function and gain of function, respectively. Other susceptibility loci point to genes involved in inflammation and airway remodeling.

UK10K haplotype resources. This yields a total of 96 million imputed variants, however only 27 million variants imputed using the HRC reference set passed the quality filters used in our study.

**Association testing and meta-analysis of disease phenotypes.** Logistic regression was used to test for association between variants and disease, assuming a multiplicative model, treating disease status as the response and expected genotype counts from imputation as covariates. We used LD score regression[12] to account for inflation in test statistics due to cryptic relatedness and stratification. We chose to include related individuals in the analysis as removing related individuals in the heavily genotyped Icelandic population would lead to the removal of most participants. Including related individuals while accounting for genomic control has proven to be a robust method[9,57,58] that does not create false positives as observed in the QQ plots (Supplementary Fig. 6; especially stratified on frequency). Software developed at deCODE genetics[57] was used to test for association in both populations. Sex, age, and county of origin were included in the logistic regression for the Icelandic dataset, and sex, age and the first 40 principal components for the UK biobank dataset. Imputation information criteria was set so that variants below 0.8 were excluded from the analysis.

Variants were mapped to NCBI Build38 positions and subsequently variants in the UK biobank imputation dataset were matched to the variants in the Icelandic dataset based on allele variation. Mantel-Haenszel model[59] was used to combine the results from the different study groups in which the groups were assumed to have a common OR but were allowed to have different population frequencies for alleles and genotypes. A likelihood ratio test was used to test heterogeneity by comparing the null hypothesis of the effect being the same in all populations to the alternative hypothesis of each population having a different effect. Heterogeneity was quantified by $I^2$ statistics which lies between 0 and 100% and describes the proportion of total variation in study estimates that is due to heterogeneity.

Variants were split into five classes based on their genome annotation and associations were considered significant if the p-value in the combined dataset was below a weighted genome-wide significance threshold based on variant annotation. With 32,463,443 sequence variants being tested the weights given in Sveinbjornsson et al. were rescaled to control the family-wise error rate (FWER)[60]. The adjusted significance thresholds are $2.6 \times 10^{-7}$ for variants with high impact ($N = 8,644$), $5.1 \times 10^{-8}$ for variants with moderate impact ($N = 149,983$), $4.6 \times 10^{-9}$ for low-impact variants ($N = 2,283,889$), $2.3 \times 10^{-9}$ for other variants in DNase I hypersensitivity sites ($N = 3,913,058$) and $7.9 \times 10^{-10}$ for all other variants ($N = 26,108,038$). Approximate conditional analyses (COJO), implemented in the GCTA software[61] were performed on lead variants, defined by lowest $P$ value at each genomic region (locus), to identify possible secondary signals. LD between variants was estimated using a set of 8700 whole-genome –sequenced Icelandic individuals. The analysis was restricted to variants within 1 Mb from the index variants and that were present in both the Icelandic and the UK biobank datasets. Based on the number of variants tested within the 55 loci (excluding the HLA region, $N = 3,993,179$), we chose a conservative $P$ value threshold of $<5 \times 10^{-8}$ for secondary signals.

**Association testing and meta-analysis of eosinophil counts.** Linear mixed model implemented by BOLT-LMM[62] was used to test for association between sequence variants and eosinophil counts, assuming an additive genetic model. We assume that the quantitative measurements follow a normal distribution with a mean that depends linearly on the expected allele at the variant and a variance–covariance matrix proportional to the kinship matrix[63]. We used LD score regression[12] to account for inflation in test statistics due to cryptic relatedness and stratification. To combine the deCODE and UK biobank results, we used a fixed-effect inverse variance method based on effect estimates and standard errors[59]. We used a likelihood-ratio test to compute all $P$ values.

**Genetic correlation.** The cross-trait LD score regression method[12] was used to estimate the genetic correlation between pairs of traits using the summary statistics from the Icelandic and UK biobank datasets. Results for about 1.2 million variants, well imputed in both datasets,were used in this analysis. and pre-computed LD scores for European populations (downloaded from [https://data.broadinstitute.org/alkesgroup/LDSCORE/eur_w_ld_chr.tar.bz2]) were used for LD information. Genetic correlation between Icelandic GWAS summary statistic was calculated for one trait and the UK biobank GWAS summary statistic for the other traits, and the vice versa, and those results subsequently meta-analyzed, to avoid bias due to overlapping samples. For AD, we also calculated genetic correlation based on the external meta-analysis dataset downloaded from the GRASP database [https://grasp.nhlbi.nih.gov/FullResults.aspx].

**Cis-eQTL analysis.** Cis-eQTL effects were analyzed using RNA sequencing data from Icelandic samples. The generation of poly(A) + cDNA sequencing libraries, RNA sequencing, and data processing were carried out as described before[64,65]. Two tissue types were available for this analysis: whole blood and adipose tissue. In total, whole blood from 7,007 individuals and adipose tissue from 686 individuals were used. To estimate the association between sequence variant and gene expression, a generalized linear regression assuming additive genetic effect was

used on rank-transformed gene expression estimates. GTEx eQTL´s for relevant tissue types (whole-blood, lung, esophagus mucosa, spleen, lymphocytes and fibroblasts) were intersected with lead GWAS association variants, and those in LD ($r^2 > 0.80$) with the lead variant. We then report eQTLs where the top eQTL signal is in LD with lead GWAS association variants.

**Enhancer and transcription start site analysis.** Variants in linkage disequilibrium (LD) with the lead variants at the previously unreported loci were identified on the basis of in-house genotype data using $r^2 > 0.8$ for pairwise comparison of the nearest 100,000 variants to define an LD class. These variants were then annotated, see results in Supplementary Data 9. Chromatin states for CD4 + T-cells were obtained through ChromHMM analyses of available epigenome data from NIH Roadmap Epigenome Mapping Consortium for 11 histone marks analyzed by ChIP-seq, together with open chromatin regions analysed by DNase-seq, integrated into 25 discrete chromatin states[66] downloaded from [http://www.roadmapepigenomics.org]. The association variants were then annotated for chromatin states involving enhancers (EnhA1, A2, W1, or W2) or DHS sites (DNA hypersensitivity sites, reflecting open chromatin configuration). To identify enhancer-gene targets, we made use of the joint effect of multiple enhancers (JEME) resource[67]. Variants within or proximal (±25 bp) to transcriptional start sites were annotated on the basis of CAGE sequencing data derived from the Fantom5 project[68] downloaded from [fantom.gsc.riken.jp/5/data]. 3′UTR variants in predicted miRNA target sites were annotated by making use of data derived from Targetscan v7.2[69] targetscan.org.

**Generation of CD30 variants.** Full-length CD30 cDNA (NM_001243) in pCMV6-Entry Myc-DDK tagged mammalian expression vector was obtained from Origene (RC219819). Cys273Tyr mutant plasmid was generated using Q5 Site-directed mutagenesis kit (New England BioLabs, E0554S) with mutagenesis primers F-5′-AAGACGCCCATATGCATGGAAC-3′ and R-5′-CTCCACAAGGTCAT CTCG-3′ (Supplementary Table 4). Plasmids were transformed into NEB Stable Competent E.coli (New England BioLabs, C3040H) and spread on LB agar plates containing 25 µg/ml kanamycin. Colonies were expanded in LB medium containing 25 µg/ml kanamycin. Plasmids were purified using Qiagen plasmid maxi kit (Qiagen, 12163), following the manufacturer's protocol. The sequences of WT and Cys273Tyr plasmids were confirmed by Sanger sequencing.

**Over expression of CD30 and CD30 Cys273Tyr in Hela cells.** One day prior to transfection $0.1 \times 10^6$/ml Hela cells (Public Health England 93021013) were seeded in DMEM medium (ThermoFisher 11995-065) supplemented with 10% fetal bovine serum (ThermoFisher 10500-064) and 50 units/ml penicillin and 50 µg/ml streptomycin (ThermoFisher 15070-063). Cells were incubated at 37 °C with 5% $CO_2$ in a humidified incubator.

After 24 hours cells were transfected using FuGENE®HD reagent (Promega E2312) following manufacture's protocol. In short DNA was diluted in OptiMem medium (ThermoFisher 31985-047) and FuGENE®HD reagent was added at a FuGENE®HD reagent: DNA ratio 3:1.

48 hours after transfection media was removed from cells and they were washed once with PBS. Cells were dislodged from wells by incubation for 5 min at 37 °C using PBS + 2 µM EDTA. Cells were collected and spun down at $300 \times g$ for 5 min, resuspended in 1 ml PBS + 2% FBS and spun down again. Cell pellet was snap frozen and kept in −80 °C for subsequent lysis preparation.

**Hela lysis preparation.** Cells were lysed using 100 µl of RIPA buffer with 1:100 Halt protease and inhibitor cocktail (Thermo Scientific 78430). Lysates were kept on ice for 10 min and subsequently spun down at 4 °C for 15 min at $14,000 \times g$. Total protein concentration in lysates was estimated using the Pierce BCA protein assay kit (ThermoFisher 23227).

**EBV cell culture.** EBV transformed lymphoblasts from homozygous carrier and non-carrier of Cys273Tyr were cultured in RPMI1640 (ThermoFisher 61870-036) supplemented with 10% fetal bovine serum (ThermoFisher 10500-064), 50 units/mL penicillin, 50 µg/mL streptomycin (ThermoFisher 15070-063) and 20 mM HEPES (ThermoFisher 15630-056). Cells were incubated at 37 °C and 5% $CO_2$ in a humidified incubator. The cells were seeded at $5 \times 10^5$ cells/ml in fresh media 2 days prior to harvesting.

**Simple Western by size quantification of CD30 signal in Hela.** Total of 0.05 mg/ml Hela lysate was run in WES from Protein Simple following manufacture's protocol using WES 12-230 kDa separation module (Biotechne SM-W002-1). Antibody used: Purified mouse anti human CD30 (clone BerH8) (BD Biosciences cat: 555827) 1:100 dilution. Precursor CD30 signal was quantified as a 143 kDa peak and mature CD30 signal as a 207-kDa peak. As a normalization for total protein we used WES Biotin detection module (Biotechne DM-004) and followed manufacture's protocol. CD30 precursor and mature signals were normalized to total protein.

**Flow cytometry analysis on Hela, EBV, and stimulated PBMCs**. EBV cells were harvested, counted and diluted to $1 \times 10^6$ cell/ml plated in a 96 well V bottom plate. Cells were then stained with primary antibodies against PE/Cy7 anti-human CD30 (Biolegend 333918), clone BY88 reported to bind at similar amino acid residues as Ber-H6[70] that has been shown to bind epitope at the N-terminus of the peptide chain up to amino acid residue 93[71] thus unaffected by a possible effect of our variant at amino acid 273. The cells were analyzed for expression of CD30 by FACS (gating strategies shown in Supplementary Fig. 7). Cryopreserved PBMC were thawed and and plated for culture in RPMI1640 (ThermoFisher 61870-036) supplemented with 10% fetal bovine serum (ThermoFisher 10500-064), 50 units/mL penicillin, 50 μg/mL streptomycin (ThermoFisher 15070-063), 20 mM HEPES (ThermoFisher 15630056) and 30 IU/ml IL-2 (R&D Systems AFL202) in a 24-well TCR plate and rested overnight. Cells were then stimulated for 48 hours with anti CD3/CD28 dynabeads (ThermoFisher 11131D). At 48 hours cells were harvested and stained with directly PE conjugated antibody (Biolegend 333906clone: BY88)) and analyzed for expression of CD30 by FACS. Two-tailed Wilcoxon matched-pairs signed rank test was used to test for significant differences in cell surface expression of CD30 on PBMCs and EBVs. Two-tailed paired t-test was used to test for significant differences in cell surface expression of CD30 on HeLa cells.

**Soluble CD30 measured by ELISA**. Soluble CD30 was measured by sandwich ELISA (ThermoFisher BMS240INST) in a 1/4 dilution and according to manufacturer's instructions. Two-tailed Wilcoxon matched-pairs signed rank test was used to test for significant differences in levels of sCD30 from PBMC cultures. Two-tailed paired t-test was used to test for significant differences in levels of sCD30 from HeLa cell cultures.

**Pathway analysis**. We used DEPICT[49] to (1) prioritize candidate causal genes at associated loci and (2) highlight enriched pathways where genes at associated loci are highly expressed. DEPICT uses gene expression data derived from a panel of 77,840 mRNA expression arrays together with 14,461 existing gene sets based on molecular pathways derived from experimentally verified protein–protein interactions[72], genotype–phenotype relationships from the Mouse Genetics Initiative[73], Reactome pathways[74], KEGG pathways[75], and Gene Ontology (GO) terms[76]. DEPICT reconstitutes these 14,461 gene sets by calculating for each gene the probability of membership in each gene set, based on similarities across the expression data. Using these membership probabilities and a set of trait-associated loci, DEPICT tests whether any of the 14,461 reconstituted gene sets are enriched for genes at the trait-associated loci, and prioritizes genes that share predicted functions with genes at other trait-associated loci. We ran DEPICT using all 88 asthma-associated variants.

**Reporting summary**. Further information on research design is available in the Nature Research Reporting Summary linked to this article.

## Data availability

The sequence variants from the Icelandic population whole-genome sequence data have been deposited at the European Variant Archive under accession code PRJEB15197. The GWAS summary statistics are available at [https://www.decode.com/summarydata].
The authors declare that the data supporting the findings of this study are available within the article, its Supplementary Information file, and upon reasonable request. The source data underlying Figs. 1–4 and supplementary Figs. 1–5 are provided as Source Data file.

## Code availability

We used publicly available software (URLs listed below) in conjunction with the above described algorithms in the sequencing processing pipeline (Whole-genome sequencing, Association testing, RNA-seq mapping and analysis): BWA 0.7.10 mem [https://github.com/lh3/bwa]; GenomeAnalysisTKLite 2.3.9 [https://github.com/broadgsa/gatk/]; Picard tools 1.117 [https://broadinstitute.github.io/picard/]; SAMtools 1.3 [http://samtools.github.io/]; Bedtools v2.25.0-76-g5e7c696z [https://github.com/arq5x/bedtools2/]; Variant Effect Predictor [https://github.com/Ensembl/ensembl-vep]; BOLT-LMM [https://data.broadinstitute.org/alkesgroup/BOLT-LMM/downloads/]; GTEX data [http://www.gtexportal.org]; GWAS catalog [https://www.ebi.ac.uk/gwas/home]; Uniprot [https://www.uniprot.org]; Roadmap epigenomics project [http://www.roadmapepigenomics.org]; Fantom5 [fantom.gsc.riken.jp/5/data]

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

## Acknowledgements

We thank the individuals who participated in this study and the staff at the Icelandic Patient Recruitment Center and the deCODE genetics core facilities. Further to all our colleagues who contributed to the data collection and phenotypic characterization of clinical samples as well as to the genotyping and analysis of the whole-genome association data. This research has been conducted using the UK biobank Resource under Application Number '24711'.

## Author contributions

T.A.O., G.L.N., D.F.G., P.S., U.T., G.T., I.J. and K.S. designed the study and interpreted the results. T.A.O., U.S.B., G.I.E., D.G., T.G., D.L., B.R.L., P.T.O., I.O., O.S. and I.J. carried out the subject ascertainment and recruitment. T.A.O., O.A.S., S.B., J.K.S., E.V.I., B.V.H., G.H.H., G.M., K.N., T.J., G.S., V.T., D.F.G., P.S., U.T., G.T. and I.J. performed the sequencing, genotyping and expression analyses. T.A.O., F.T., K.B., A.B.A., A.M.K., S.G., U.T., G.L.N. and I.J planned and performed the functional lab work. T.A.O., O.A.S., S.B., J.K.S., E.V.I., B.V.H., A.G., G.H.H., G.M., K.N., T.J., L.S., G.S., V.T., D.F.G., P.S., U.T., G.T. and I.J. performed the statistical and bioinformatics analyses. T.A.O., O.A.S., D.F.G., P.S., U.T., I.J. and K.S. drafted the manuscript. All authors contributed to the final version of the paper.

## Competing interests

Authors affiliated with deCODE genetics/Amgen Inc. declare competing interests as employees. The remaining authors declare no competing interests.
