## [Peer Review File · Nature Communications]

Reviewers' Comments:

Reviewer #1:

Remarks to the Author:

In the manuscript "Eighty-nine asthma risk variants highlight the role of T cell regulation, inflammation and airway remodeling in asthma pathogenesis" Olafsdottir et al. used two large study populations from deCODE AND UK Biobank in (UKB) to conduct a meta-GWAS on asthma. They report 89 independent signals at 56 loci of which 21 signals at 17 loci represented novel asthma loci. The respective candidate genes were identified based on functional annotations, eQTL data, and regulatory sites covered by the associated SNPs. In addition, for two variants genotype-specific expression data are generated. The reported genes are involved in T-cell signaling, innate immunity and airway remodeling.

Major comment

The manuscript is well written and the methods are sound. Major assets of the study are the large data sets, providing power for discovery of novel loci. The phenotype definitions are appropriate and the replication of a large number of known asthma/allergy loci points to the validity of the approach. Nonetheless several points need to be clarified.

For imputation, the authors used different methods and different reference sequences. They should comment on this and whether it might have introduced a bias. They do not report any QQ plots, neither for the individual populations nor for the combined analysis. Please provide the plots and information on genomic inflation factors.

Essentially, genome-wide significant association signals are derived from UKB, the majority of loci does not replicate in the Icelandic population. Please comment on this.

What is the advantage of the Mantel-Haenszel model used for meta analysis. It assumes homogeneous ORs in the study populations but some loci show heterogeneity. How does this affect the validity of the results? Are the results different from the results calculated with METAL?

Additional points

On page 3, how was the conditional analysis performed? If conditioning on the lead variant identified more than one additional genome-wide significant signal, how was the number of independent signals determined? This should be explained in the methods section in more detail.

On page 5, replace IL21 with IL21R

On page 7, the effect of the IL33 variant was not only greater in EOA but also in LOA than in AA. Accordingly, this might be a locus specific for non-allergic asthma.

On page 7, the correlation ρ and r^2 values for asthma vs allergic rhinitis and allergic dermatitis do not agree with supplementary figure 5a and 5b respectively.

On page 20 (methods), please specify the Biologend antibodies.

In figure 2b and c, please explain the number of dots.

In figure 3, please explain the sequences in more detail. Why are some bases on top of the 5' sequence or below the 3' sequence? Why do the red sequences on the left and on the right differ?

In table 1, please check the overlap with Ferreira et al. (Nat Genet. 2017 Dec;49(12):1752-1757) and with Shrine et al. (Lancet Respir Med. 2019 Jan;7(1):20-34).

On page 16, the p-value for the secondary signal should not be based on the number of loci ($n=56$). Please change this statement. The applied threshold is correct.

In supplementary fig 1, please show the whole size range and also the loading control.

In supplementary fig 4, delete "Ref" in the header or add the reference.

In supplementary fig 5, the scaling of both axes should be identical.

In supplementary tab 1, what was conditioned on in the combined analysis? Comment on heterogeneity of effects? Which threshold was used and why?

At the IL13 locus, would conditioning on rs848 eliminate the other 2 signals? Could this be the causal variant?

In supplementary tab 2, please add the risk allele for the previously reported SNPs in order to compare the effect direction.

In supplementary tab 3, what was conditioned on?

In all tables ordered according to the SNP positions, please check the order of SNPs.

In supplementary tab 5, the confidence interval is missing, rsq values are missing, p-values from Ferreira et al are missing.

Why are p-values from Ferreira et al different for the same SNP (e.g. rs73205303)?

Novel loci are not associated in the Icelandic population, please comment.

In supplementary tabs 7 to 9, why are only novel SNPs included? Since previous studies performed different functional analyses, the authors should report the results for all independent signals.

In supplementary tab 8, please report the effect alleles. Why is whole blood from GTEx not included. It would be interesting to have a comparison of the results with the deCODE data (suppl tab 6).

In supplementary tab 9, why were transcription factor binding sites not included?

In supplementary tab 10, for rs6926894, the OR of UKB is reported instead of deCODE, the CI is missing.

In supplementary tab 12, the number of UKB asthma cases does not agree with page 2, why?

Reviewer #2:

Remarks to the Author:

The authors carried out the largest GWAS meta-analysis of asthma to date with over 69,000 cases and over 700,000 controls. In total 89 independent asthma variants were identified. Of these:

- 48 signals in 24 known asthma loci: 24 new/24 previously reported
- 20 signals in 15 known 'combined allergic phenotype (asthma/hay fever/eczema)' loci
- 21 signals in 17 new asthma loci

They also quantify the correlation of the effects with other asthma and allergic phenotypes and perform functional and eQTL analyses to investigate the implicated genes and their functional roles.

This work is novel in that it is the largest ever meta-analysis; the major claims are the discovery of 21 new independent asthma loci, of which 18 are common (MAF \geq 5%) and 3 uncommon variants, 2 of the uncommon variants show a 'small' protective effect for asthma. Most of the genes implicated in the reported associations with asthma are involved in known T cell regulation, inflammation and airway remodeling.

I praise the authors for the detail of the data reported in the supplementary tables. These data will be essential for other researchers to be able to use the results of this meta-analysis. For instance, complete GWAS summary statistics are needed to perform Mendelian Randomization studies that I'm sure will follow these findings.

Comments

1. In the main text the authors state that the asthma phenotype used for this meta-analysis includes both definitions: doctor ever diagnose asthma and/or self-reported asthma. It is unclear whether the self-reported asthma is self-reported doctor diagnosis of asthma. Can the authors comment on the validity of the cases? What proportion of cases were defined from a doctor diagnoses, self-reported asthma and self-reported doctor diagnosis? Could there be an over-reporting of asthma cases (false positives) and what would the implications be?

2. Main results are described in page 3. It took this reviewer several reads to have a clear understanding of the old/new findings. A suggested opening sentence would be: "Conditional analysis yield 89 independent signals: 48 of these signals in 24 known asthma loci (50% of the signals are new), 20 in 15 known combined (asthma-eczema-rhinitis) phenotype loci and 21 new signals in 17 new asthma loci." And then expand from here..

3. The authors don't seem to provide a description of the studied populations. How do the Iceland and UK populations compare in terms of general demographics?

4. Could the author comment on the magnitude and clinical implications of the protective effects? Comb OR (95%CI) = 0.82 (0.78, 0.87) for variant near TNFRSF8, Comb OR (95%CI) =0.89 (0.86, 0.92) for variant near TGFBR1.

5. Table 1. Confidence intervals are not provided. Can the authors add the corresponding 95 % CIs (as in Suppl Table 1) or the standard errors so the effects can be fully interpreted together with the p-values?

6. The authors have investigated the effect of the asthma variants on early onset (first diagnosis ≤ 17 years) and late onset (first diagnosis > 17 years) asthma and found that for some variants the effect was stronger on early onset asthma compared to late onset asthma. Do the authors think that these results support the notion that early and late onset asthma have partly distinct genetic architectures? And if so, what implications can be drawn?

Reviewer #3:

Remarks to the Author:

The findings might be interesting but the experimental part that meant to support genetic findings is weak and lacks appropriate quality controls.

Figure 2 b and c are underpowered. There is also a lack of evaluation of soluble CD30 in supernatants from Hela cells transfected with different CD30 variants.

Figure 2 d and e. There is no mention of statistical test used to test difference in CD30 surface expression and soluble levels in d and e. Authors linked heterozygous carriers with age and gender matched non carrier controls suggesting that paired t test was used. I am not convinced that this is appropriate. The data in d might be skew by two pairs.

The representative flow cytometry plots of CD30 surface expression for c-d are required.

Figure 3. The conclusion that rs41283642 increases TGFb expression is not supported experimentally. Evaluation of surface TGFBR1 expression in people with different variants is recommended. Also, in order to claim that rs41283642 increases TGFb expression because of a

lack of negative regulation by microRNA needs to be tested experimentally.

Supplementary Figure 1 does not have a loading control

Supplementary Figure 2 is underpowered (n=3)

Reviewers' comments:

Reviewer #1 (Remarks to the Author):

In the manuscript “Eighty-nine asthma risk variants highlight the role of T cell regulation, inflammation and airway remodeling in asthma pathogenesis” Olafsdottir et al. used two large study populations from deCODE AND UK Biobank in (UKB) to conduct a meta-GWAS on asthma. They report 89 independent signals at 56 loci of which 21 signals at 17 loci represented novel asthma loci. The respective candidate genes were identified based on functional annotations, eQTL data, and regulatory sites covered by the associated SNPs. In addition, for two variants genotype-specific expression data are generated. The reported genes are involved in T-cell signaling, innate immunity and airway remodeling.

Major comment

The manuscript is well written and the methods are sound. Major assets of the study are the large data sets, providing power for discovery of novel loci. The phenotype definitions are appropriate and the replication of a large number of known asthma/allergy loci points to the validity of the approach. Nonetheless several points need to be clarified.

1. For imputation, the authors used different methods and different reference sequences. They should comment on this and whether it might have introduced a bias. They do not report any QQ plots, neither for the individual populations nor for the combined analysis. Please provide the plots and information on genomic inflation factors.

Response:

Both datasets are imputed based on very large reference sets, the Icelandic population specific and the HRC panel, for UK biobank, that includes a large number of UK individuals. This results in high quality imputation for both datasets, better than could be obtained by using for both populations common publicly available reference set, such as the 1000 genomes training set. There can still be difference in the imputation quality for individual markers between the datasets, primarily due to the different chip genotype platforms used rather than the different reference set used in the imputation. This should, however, not lead to false positive results as low imputation quality, unless correlated with affection status, would simply reduce the power to detect association. All novel associated variants we report have imputation info > 0.87 in both datasets. We have added the QQ plot (shown below) as supplementary figure 6 and information on the genomic inflation factors, estimated using LD-score regression (Bulik-Sullivan, B.K., Nat Gen, 2015) has been added to the materials and methods chapter as described in next response. Note that the large inflation factors for the Icelandic dataset are due to the relatedness of the individuals in the cohort (the Icelandic dataset includes majority of the Icelandic population).

Supplementary Figure 6. Quantile-quantile plot (QQ-plot) showing chi-square statistics corrected for genomic control and showing variants found in UK and UK cohorts for the GWAS of Asthma in Iceland (red) the UK (blue) and meta-analysis of the two cohorts (black). The red diagonal line represents expected distribution assuming no inflation of the chi-square statistics.

2. Essentially, genome-wide significant association signals are derived from UKB, the majority of loci does not replicate in the Icelandic population. Please comment on this.

Response: We note that the effective sample size is four times bigger for the UK Biobank samples than for the Icelandic samples, hence most of the power in the analysis comes from the UK samples. We have added a description of this in the Material and Methods section page 15 "In total we had 16,247 asthma cases and 346,486 controls from Iceland and 52,942 cases and 355,713 controls from the UK Biobank. Estimating the effective number of cases as $2 \times N_a \times N_c / (N_a + N_c) / GC$, where N_a and N_c are the number of cases and controls, respectively, and GC is the genomic adjustment factor, estimated using LD-score regression (Bulik-Sullivan, B.K., Nat Gen, 2015), we can calculate the effective sample size for Iceland to be $N_{eff}=22,689$ and UK Biobabank to be $N_{eff}=84,396$, $GC=1.368$ in Iceland and $GC=1.092$ in UK Biobank". We further note there is no significant heterogeneity ($P < 0.05/88 = 5.62E-04$) in effect sizes between the two sample sets for vast majority of the variants, hence the lack of significance in the Icelandic data simply reflects a lower power to detect association. We have added a sentence on this on page 3 "Based on number of cases and controls, we note that the effective sample size is four times bigger for the UK samples ($N=84,396$) than the Icelandic ($N=22,689$), hence most of the power comes from the UK samples. We note that only 4 out of the 88 variants show significant heterogeneity based on the number of variants tested ($P < 0.05/8 = 5.62 \times 10^{-4}$) and 28 show nominal heterogeneity ($P < 0.05$) in effect sizes between the two sample sets demonstrating a good consistency of effect for vast majority of the variants in the Icelandic samples. Further, 85 out of the 88 variants ($P < 1 \times 10^{-16}$) have effect size in the same direction in the two cohorts and 44 are at least nominally significant in Iceland showing a good replication of effects (Supplementary Table 1)"

3. What is the advantage of the Mantel-Haenszel model used for meta analysis. It assumes homogeneous ORs in the study populations but some loci show heterogeneity. How does this affect the validity of the results? Are the results different from the results calculated with METAL?

Response: The primary justification for using a fixed effect meta-analysis is that in majority of cases we do not detect a significant heterogeneity in the effect estimates, i.e. only 4 out of 88 variants have different effect estimates given the number of variants tested ($P < 0.05/88 = 5.62E-04$). However, to investigate whether this assumption affects our result we have re-evaluated all the asthma variants using a sample size weighted meta-analysis using METAL. All of the lead variants at the reported 56 loci are GWAS significant even when we do not assume homogeneous OR in the study populations. The choice of Mantel-Haenszel model does therefore not affect the validity of the results. We have added the P values from the METAL analysis in supplementary Table 1.

Additional points

4. *On page 3, how was the conditional analysis performed? If conditioning on the lead variant identified more than one additional genome-wide significant signal, how was the number of independent signals determined? This should be explained in the methods section in more detail.*

Response: We have added more information on the conditional analysis in the material and methods section page 17 „Approximate conditional analyses (COJO), implemented in the GCTA software⁵¹ were performed on lead variants, defined by lowest P value at each genomic region (locus), to identify possible secondary signals. LD between variants was estimated using a set of 8700 whole-genome –sequenced Icelanders. The analysis was restricted to variants within 1 Mb from the index variants that were present in both the Icelandic and the UK datasets. Based on the number of variants tested within the 55 loci (excluding the HLA region, N=3,993,179), we chose a conservative P value threshold of $< 5 \times 10^{-8}$ for secondary signals.” In the review process we realized that for some of the loci a P value cut-off of $< 1 \times 10^{-7}$ was used. Re-evaluating those secondary signals with the cut-off $< 5 \times 10^{-8}$ did not change anything except for one secondary signal (rs3781094) at the GATA3 locus that did not reach this threshold and was therefore removed from supplementary table 1. The total number of variants reaching our threshold therefore, changed from 89 to 88.

5. *On page 5, replace IL21 with IL21R*

Response: This has been corrected

6. *On page 7, the effect of the IL33 variant was not only greater in EOA but also in LOA than in AA. Accordingly, this might be a locus specific for non-allergic asthma.*

Response: We agree that in our data the effect of the IL33 variant does not seem to be driven by allergic asthma. It has the greatest effect on EOA, followed by LOA whereas smallest effect, that does not reach nominal significance, on AA. However, we lack information on AA, especially in the UK samples where only 542 (1% of all asthma cases) have this diagnosis and there is overlap both with the early and late onset asthma cases. Therefore, this warrants further investigation. We have therefore, added this to page 7-8 “Further, the effect of IL33 (rs149045797) variant was largest on EOA (OR=0.53, P=3.8E-09) followed by LOA (OR=0.71, P=5.1E-09) with smallest effect on AA (OR=0.93, P=0.70; Figure 4 and supplementary table 13). This variant is fully correlated ($r^2=1$) with a loss of function variant (rs146597587) previously reported to associate with reduced IL33 mRNA expression and asthma risk and in line with the results reported here, the largest effect was observed in young children that were hospitalized at least 4 times before the age of 6 years due to asthma exacerbation (OR=0.24, P=0.04)³³. However, 58% of those young children had atopic diagnosis before 6 years of age⁴⁰ indicating that the IL33 association is driven by severe asthma rather than non-AA exclusively. We note that we lack information on AA for a large part of our samples, especially in the

UK samples where only 542 (1% of all asthma cases) have this diagnosis and there is overlap both with the early and late onset asthma cases. Therefore, the IL33 association with asthma severity in AA and non-AA warrants further investigation.”

7. On page 7, the correlation p and rsq values for asthma vs allergic rhinitis and allergic dermatitis do not agree with supplementary figure 5a and 5b respectively.

Response: This has been corrected.

8. On page 20 (methods), please specify the Biologend antibodies.

Response: This has been added

9. In figure 2b and c, please explain the number of dots.

Response: The following has been added in the legend to Figure 2 “ The dots in plots b-d represent individual experiments. The dots in plots e and f represent individual donors. Lines in panel b-d indicate median level”.

10. In figure 3, please explain the sequences in more detail. Why are some bases on top of the 5'-sequence or below the 3' sequence? Why do the red sequences on the left and on the right differ?

Response: Although the question is not entirely clear, we assume that the reviewer is referring to the left-hand side of the figure showing the primary miR-142. This figure shows the folded transcript hence, the sequence is shown at the RNA level. The figure legend additionally clearly states that this is the „transcript“. As to the latter part of the question, there was a missing „c“ on the 3p sequence for which we apologize and we have now corrected this mistake.

11. In table 1, please check the overlap with Ferreira et al. (Nat Genet. 2017 Dec;49(12):1752-1757) and with Shrine et al. (Lancet Respir Med. 2019 Jan;7(1):20-34).

Response: We have carefully checked the overlap with Ferreira et al which is reported in supplementary Table 5. The Shrine paper was not published when we submitted our manuscript but we observe that two of the novel signals claimed in their publication are represented at 2 of the novel loci reported by us. rs12788104 (chr11:1129831) reported by us at the MUC2/MUC5AC locus has low correlation with the variant reported in Shrine et al as a novel variant at MUC2/MUC5AC locus rs11603634 (chr11:1142570; $r^2=0.35$). The rs11603634 does not reach significance in our study ($P=1.20E-05$, effect=1.03). Our secondary signal at the KIAA1109 locus (rs17454584) is highly correlated with the novel signal Shrine reported at this locus rs72687036 ($r^2=0.95$) whereas our primary signal rs45613035 is not ($r^2=0.02$). The variant at the GATA3 locus does not correlate with any of the 7 independent variants we report at this locus. The discrepancy between the two studies could be explained by the different selection criteria where we included all asthma cases whereas they only selected moderate-severe asthma subjects from UK Biobank under the following criteria; Asthma – doctor diagnosis (data field 6152), Exclude doctor diagnosed Emphysema/chronic bronchitis (data field 6152) and moderate-severe asthma based on British Thoracic Society (BTS) guidelines (2014) and the British National Formulary (September 2015) (data field 20003).

We have marked these variants in Table 1, and added the following text below the table “ + loci reported in Shrine et al.⁴⁶ while this paper was in review. The novel variant reported by Shrine et al.

at MUC5AC/MUC2 locus (rs11603634) is weakly correlated with a variant we report here at this locus (rs12788104; $r^2=0.35$) and the reported variant at KIAA1109 locus (rs72687036) is highly correlated with a secondary signal (rs17454584) we report here at that locus. The discrepancy between the two studies could be explained by the different definitions of the asthma phenotypes; we included all asthma cases whereas they only selected moderate-severe asthma cases.”

12. *On page 16, the p-value for the secondary signal should not be based on the number of loci (n=56). Please change this statement. The applied threshold is correct.*

Response: The reviewer is correct that it is based on the number of variants but not loci. We have adjusted the text on page 17 related to this “Based on the number of variants tested within the 55 loci (excluding the HLA region, N=3,993,179, we chose a conservative P value threshold of $<5 \times 10^{-8}$ for secondary signals.”

13. *In supplementary fig 1, please show the whole size range and also the loading control.*

Response: In combination with an increased number of experiments ($n=6$ now instead of $n=3$ before), we have added the relevant loading controls to Supplementary Figure 1. As we are using WES (Protein simple, BioTechne) for our analyses instead of conventional Western blot, CD30 and total protein quantifications were run in separate wells from the same lysis preparation. For the normalization calculation we followed instructions from Protein simple for the total protein module (Biotechne DM-TP01). Precursor and mature CD30 were normalized to total protein.

14. *In supplementary fig 4, delete “Ref” in the header or add the reference.*

Response: This has been corrected by adding the reference (Ferreira et al. 2017). Furthermore, the secondary header was removed “Reported allergy variants and risk of asthma”

15. *In supplementary fig 5, the scaling of both axes should be identical.*

Response: This has been changed in a new supplementary fig 5.

16. *In supplementary tab 1, what was conditioned on in the combined analysis? Comment on heterogeneity of effects? Which threshold was used and why?*

At the IL13 locus, would conditioning on rs848 eliminate the other 2 signals? Could this be the causal variant?

Response: More information regarding the conditional analysis has been added to the material and methods section as described in a previous response regarding the conditional analysis. In the conditional analysis we start with conditioning on the most significant signal at this locus (in the case of IL13 locus it was the rs848) and continue with step-wise regression using all variants reaching the $P < 5 \times 10^{-8}$ cut-off. In the end, variants reaching the P value cutoff are then used to condition on the top signal so that all variants reaching a given P value cut-off are used to condition on. That means that the rs848 did not reach the $P < 5 \times 10^{-8}$ cut-off after conditioning on rs3749833 and rs60156285 whereas those two signals reached the $P < 5 \times 10^{-8}$ cut-off (and in fact got stronger P values) when conditioned on rs848.

17. In supplementary tab 2, please add the risk allele for the previously reported SNPs in order to compare the effect direction.

Response: The table has been modified to report the risk allele based on information found in GWAS catalog.

18. In supplementary tab 3, what was conditioned on?

Response: This is the same conditional analysis as reported in supplementary table 1 and described above. A few inconsistencies were observed between the P values in the two tables and this has been corrected.

19. In all tables ordered according to the SNP positions, please check the order of SNPs.

Response: This has been checked and corrected.

20. In supplementary tab 5, the confidence interval is missing, rsq values are missing, p-values from Ferreira et al are missing.

Response: The CI, risk allele, p values and OR from the Ferreira paper have been added to supplementary table 5. The r^2 values were previously given in column Q but are now after the addition of columns in column W. This has been clarified by adding this header on top of column S " r^2 between Asthma lead SNP and SNP reported in Ferreira". The conditional analysis has been moved from the last column to column N.

21. Why are p-values from Ferreira et al different for the same SNP (e.g. rs73205303)?

Novel loci are not associated in the Icelandic population, please comment.

Response: We think this might be a misunderstanding due to the lack of a descriptive header for the r^2 column in the previous version. Indeed the p-value for the same SNP is always the same in Ferreira but we test if they correlate with all the significant markers we find at the previously reported Ferreira loci. For example, rs73205303 reported in Ferreira, is highly correlated ($r^2=0.97$) with our lead marker at that locus (rs11088309) but not ($r^2=0.02$) with the secondary signal (rs8129030) we report at that locus. With regard to the lack of significant association in the Icelandic dataset we refer to response no 2.

22. In supplementary tabs 7 to 9, why are only novel SNPs included? Since previous studies performed different functional analyses, the authors should report the results for all independent signals.

Response: While we agree with the reviewer that this would be interesting we also note that this requires a lot of work and the previously reported variants are not the main scope of this manuscript. Furthermore, the information provided in supplementary tables 7 to 9 can all be retrieved from publicly available databases and is therefore available for interested readers to look into for all variants presented in this manuscript. On the other hand we screened all the 88 variants for our in-house blood and adipose eQTL data as presented in supplementary table 6.

23. In supplementary tab 8, please report the effect alleles. Why is whole blood from GTEx not

included. It would be interesting to have a comparison of the results with the deCODE data (suppl tab 6).

Response: The effect alleles have been added. The deCODE whole blood data is based on much larger dataset (n=7007) than GTEx whole blood samples (n=369 genotyped individuals) and therefore the deCODE data is much more informative. Additionally, a comparison of two datasets is confounded by multiple factors, i.e. sequencing depth, RNA extraction and library preparation protocols, which is beyond the scope of our current study.

24. In supplementary tab 9, why were transcription factor binding sites not included?

Response: We have now included transcription factor binding sites in supplementary table 9. We identified minor alleles that disrupt a binding site for a given TF, using motifs from HOCOMOCO v11, and report binding sites supported by a chip-seq peak found in GTRD database.

25. In supplementary tab 10, for rs6926894, the OR of UKB is reported instead of deCODE, the CI is missing.

Response: This has been corrected and CI added.

26. In supplementary tab 12, the number of UKB asthma cases does not agree with page 2, why?

Response: Unfortunately, we do not have information on age of onset or allergic asthma for all cases, in either the UKB or Icelandic samples.

Reviewer #2 (Remarks to the Author):

The authors carried out the largest GWAS meta-analysis of asthma to date with over 69,000 cases and over 700,000 controls. In total 89 independent asthma variants were identified. Of these:

- 48 signals in 24 known asthma loci: 24 new/24 previously reported**
- 20 signals in 15 known 'combined allergic phenotype (asthma/hay fever/eczema)' loci**
- 21 signals in 17 new asthma loci**

They also quantify the correlation of the effects with other asthma and allergic phenotypes and perform functional and eQTL analyses to investigate the implicated genes and their functional roles.

This work is novel in that it is the largest ever meta-analysis; the major claims are the discovery of 21 new independent asthma loci, of which 18 are common (MAF \geq 5%) and 3 uncommon variants, 2 of the uncommon variants show a 'small' protective effect for asthma. Most of the genes implicated in the reported associations with asthma are involved in known T cell regulation, inflammation and airway remodeling.

I praise the authors for the detail of the data reported in the supplementary tables. These data will be essential for other researchers to be able to use the results of this meta-analysis. For instance, complete GWAS summary statistics are needed to perform Mendelian Randomization studies that I'm sure will follow these findings.

Comments

1. In the main text the authors state that the asthma phenotype used for this meta-analysis includes both definitions: doctor ever diagnose asthma and/or self-reported asthma. It is unclear whether the self-reported asthma is self-reported doctor diagnosis of asthma.

Can the authors comment on the validity of the cases? What proportion of cases were defined from a doctor diagnoses, self-reported asthma and self-reported doctor diagnosis? Could there be an over-reporting of asthma cases (false positives) and what would the implications be?

Response: We thank the reviewer for raising this important topic. We have now added the number of cases for self-reported asthma and doctors diagnoses in a supplementary table 12 (note that there is an overlap between the two diagnosis both in Iceland (N=3224) and in the UK (N=21092). All of the self-reported asthma is self-reported doctors diagnosis as described in materials and methods “self-reported by a positive reply to the question: “Has a doctor confirmed your asthma diagnosis”. We have added this into the main text on page 2 “The asthma phenotype in both sets was based on physician diagnoses and/or self-reported doctor’s diagnosis of asthma” In supplementary table 13 we report OR and P value for all the 88 variants for ICD10 codes and self-reported cohorts separately. Similar effect size is observed for all variants and therefore, we observe no evidence for a bias between the two asthma sample sets.

2. Main results are described in page 3. It took this reviewer several reads to have a clear understanding of the old/new findings. A suggested opening sentence would be: “Conditional analysis yield 89 independent signals: 48 of these signals in 24 known asthma loci (50% of the signals are new), 20 in 15 known combined (asthma-eczema-rhinitis) phenotype loci and 21 new signals in 17 new asthma loci.” And then expand from here.

Response: We thank the reviewer for this suggestion and have modified the text on page 3 in the following way: “Of the 88 independent signals, 47 are at 24 previously reported asthma loci, 20 variants at 15 loci have previously been reported for a combined allergic phenotype⁷ and 21 variants at 17 loci are novel asthma signals.”

3. The authors don’t seem to provide a description of the studied populations. How do the Iceland and UK populations compare in terms of general demographics?

Response: Information on year of birth and sex for the two samples has been added in supplementary table 17. The age range is wider in the Icelandic cohort mean Year of birth 1965.4 (SD:26.5) in Iceland and 1951.6 (SD: 8.2) in the UK whereas the gender ratio is similar 40.6% males in Iceland and 42.5% in the UK. The different age range is explained by the way recruitment was conducted in the UK biobank where all subjects were aged between 40 and 69 at recruitment whereas there were no age restrictions in the Icelandic recruitment.

4. Could the author comment on the magnitude and clinical implications of the protective effects? Comb OR (95%CI) = 0.82 (0.78, 0.87) for variant near TNFRSF8, Comb OR (95%CI) =0.89 (0.86, 0.92) for variant near TGFBR1.

Response: *TNFRSF8* variant has the greatest protective effects of the 21 novel variants reported here. Given the heterogeneity of asthma this can be viewed as a considerable effect on asthma risk and in fact there are only four variants affecting asthma risk more than the *TNFRSF8* variant in our dataset (rs72782676 (intergenic on chr10, OR_{comb}: 0.63(0.58-0.69), rs149045797 (IL33, OR_{comb}:0.65 (0.59-0.72), rs12722502 (IL2RA, OR_{comb}: 0.80(0.77-0.85) and rs61816761 (FLG, 1.24 (1.19-1.29)). Further evaluation of the effect size of those variants in different asthma endotypes will be needed to fully evaluate the clinical implication of those variants.

We have added a sentence on this on page 4: “ The p.Cys273Tyr variant had the greatest protective effects (OR=0.82, P=8.27×10⁻¹³) of the 21 novel variants reported here and in fact there were only 4 out of the 88 asthma variants with greater effect on asthma risk; rs72782676 (intergenic on chr10, OR=0.63), rs149045797 (intron variant at *IL33* locus, OR=0.65), rs12722502 (intron variant at *IL2RA* locus, OR=0.80 and rs61816761 (stop-gained variant at *FLG* locus, OR=1.24).”

5. Table 1. Confidence intervals are not provided. Can the authors add the corresponding 95 % CIs (as in Suppl Table 1) or the standard errors so the effects can be fully interpreted together with the p-values?

Response: The 95% CI have been added into Table 1. In order to facilitate this we had to remove the column showing I² that can now only be observed in supplementary table 1.

6. The authors have investigated the effect of the asthma variants on early onset (first diagnosis ≤17 years) and late onset (first diagnosis >17 years) asthma and found that for some variants the effect was stronger on early onset asthma compared to late onset asthma. Do the authors think that these results support the notion that early and late onset asthma have partly distinct genetic architectures? And if so, what implications can be drawn?

Response: The reviewer correctly pointed out that these results were not discussed further in the manuscript.

We have added a sentence on this on page 8. “These results point to a partly distinct genetic architecture behind the three asthma phenotypes tested in our analysis in line with the clinical heterogeneity of the different asthma phenotypes. Our results show that certain sequence variants have stronger effect in EOA than LOA indicating that genetics may play a larger role in EOA.”

Reviewer #3 (Remarks to the Author):

The findings might be interesting but the experimental part that meant to support genetic findings is weak and lacks appropriate quality controls.

Comments:

1. Figure 2 b and c are underpowered.

Response: We agree with the reviewer on this point and to that end, we have now increased the number of experiments to six. Further, we optimized our WES protocol so that we could better define precursor and mature forms of the CD30 protein. The results from these six experiments are consistent with, and confirmed our previous results showing that in whole cell lysate from HeLa cells expressing the p.Cys273Tyr variant we observe higher ratio of precursor/mature CD30 protein, compared to cells expressing the WT version ($P = 4.2 \times 10^{-7}$, Fig. 2b). Further, by increasing the number of the HeLa cell experiments we observe significantly lower CD30 surface expression on p.Cys273Tyr cells than the WT ($P=7.4 \times 10^{-5}$, Fig. 2c) in line with what we observe on stimulated PBMCs. We have therefore, added information on this on page 4 in the following way: "CD30 is expressed as a precursor protein that undergoes post-translational modification, which results in the mature form of the protein^{18,19}. Lysates from cells expressing the p.Cys273Tyr variant had higher ratio of the precursor form compared to the mature form than that observed in cells expressing the WT CD30 ($P=4.2 \times 10^{-7}$ 1.2×10^{-4} ; Fig.2b and Supplementary fig. 1). Moreover we observed a lower surface expression on p.Cys273Tyr cells than WT CD30 ($P=7.4 \times 10^{-5}$, Fig. 2c)."

2. There is also a lack of evaluation of soluble CD30 in supernatants from HeLa cells transfected with different CD30 variants.

Response: To address this important point we measured the amount of soluble CD30 in the supernatants of HeLa cells expressing either WT or p.Cys273Tyr CD30. These experiments showed a markedly lower amounts of soluble CD30 generated from cells expressing the p.Cys273Tyr variant CD30 ($N = 6$, $P = 5.5 \times 10^{-4}$), consistent with our observation of lower soluble CD30 concentration in supernatants of PBMC from heterozygous carriers of p.Cys273Tyr. We have added a sentence on this on page 4: "and a significantly lower amount of sCD30 in the cell culture supernatant of cells expressing the p.Cys273Tyr variant ($P=5.5 \times 10^{-4}$, Fig. 2d)."

3. Figure 2 d and e. There is no mention of statistical test used to test difference in CD30 surface expression and soluble levels in d and e. Authors linked heterozygous carriers with age and gender matched non carrier controls suggesting that paired t test was used. I am not convinced that this is appropriate. The data in d might be skew by two pairs.

Response: For the PBMC analyses we used two-tailed Wilcoxon matched-pairs signed rank test to test for significant differences in cell surface and soluble expression between p.Cys273Tyr and WT CD30. We have added further description to this in the material and methods chapter page 21.

4. The representative flow cytometry plots of CD30 surface expression for c-d are required.

Response: We have now incorporated into Figure 2 representative flow cytometry plots (histograms) depicting the expression levels of CD30 for HeLa cells expressing WT or p.Cys273Tyr CD30, and equivalent analysis of PBMC from a non-carrier and p.Cys273Tyr heterozygous carrier.

5. Figure 3. The conclusion that rs41283642 increases TGFb expression is not supported experimentally. Evaluation of surface TGFbR1 expression in people with different variants is recommended. Also, in order to claim that rs41283642 increases TGFb expression because of a lack of negative regulation by microRNA needs to be tested experimentally.

Response: As stated in the manuscript we do observe a significantly increased ($P=7.24 \times 10^{-101}$) expression of TGFBR1 mRNA in RNA sequencing data from peripheral blood cells from heterozygous carriers of rs41283642-T located within an micro RNA binding site for miR-142-3p in the 3' UTR of TGFBR1. The increase in TGFBR1 expression is substantial or 15.2% per rs41283642-T allele demonstrating a strong link between the rs41283642-T and TGFBR1 expression. Furthermore, others (Zhe Lei et al. 2014) have shown that the microRNA miR-142-3p regulates TGFBR1 expression by binding to the 3'UTR of TGFBR1 and suppressing its expression. This was demonstrated both at the RNA and protein level. The asthma associated variant alters the binding site of miR-142-3p in the 3'UTR of TGFBR1. Collectively these data provide strong evidence that rs41283642-T leads to increased expression of TGFBR1. We have described in more detail the results presented by Zhe Lei et al, in the result section of the revised manuscript (page 5), to provide further support for the increase in expression of TGFBR1 because of lack of negative regulation by miR-142-3p:

„Expression quantitative trait locus (eQTL) analysis showed that the variant associates with 15.2% increasing TGFBR1 expression in blood per allele ($P=7.24 \times 10^{-101}$, effect=1.18 SD, which is the most significant eQTL for TGFBR1; Fig. 3a-b, Supplementary Table 6).rs41283642 T changes a recognition site for microRNA (miR) miR-142-3p (Fig. 3c). MiR-142-3p has previously been shown by others¹⁹ to bind to the 3'UTR site in TGFBR1 and repress TGFBR1 expression at both RNA and protein levels. Interestingly, miR-142-3p is one of only three miRs reported to have an increased expression in severe asthmatic lungs²⁰. Together these data suggest that rs41283642 T reduces the binding of miR-142-3p to TGFBR1 transcripts leading to increased TGFBR1 expression and through that protection against asthma“

6. Supplementary Figure 1 does not have a loading control

Response: In combination with an increased number of experiments, we have now added the relevant loading controls to Supplementary Figure 1. As we are using WES (Protein simple, BioTechne) for our analyses instead of conventional Western blot, CD30 and total protein quantifications were run in separate wells from the same lysis preparation. For the normalization calculations we followed instructions from Protein simple for the total protein module (Biotechne DM-TP01). Precursor and mature CD30 were normalized to total protein.

7. Supplementary Figure 2 is underpowered (n=3)

Response: The experiments displayed in Supplementary Figure 2 involve primary cells donated by homozygous carriers of the p.Cys273Tyr variant. We know of sixteen homozygous carriers in Iceland that are alive and we are in the unique position to be able to recruit them based on their genotype. We attempted to recruit all sixteen homozygotes and at the time of submission, we had recruited four. In the meantime, we were able to recruit two additional homozygotes. Since we are dependent on the voluntary participation of individuals we contact, we currently have no means to increase the number of samples beyond these six. The current number of samples is six, and we have updated the Supplementary Figure 2 accordingly.

Reviewers' Comments:

Reviewer #1:

Remarks to the Author:

The authors have addressed most of the comments in an accurate manner and the manuscript has improved considerably.

However, there are still some concerns regarding genomic inflation which seems to be a major problem in both studies according to Supplementary Figure 6. The authors should present the results on genomic inflation in more detail in the results section (genomic inflation factors for both studies) and comment on this in the discussion section. What is the inflation factor in UKB and what are potential reasons for genomic inflation in UK Biobank? Have previous studies reported similar inflation in UKB? If population structure/relatedness is a major concern in the Icelandic study, why have the authors not used a method that effectively controls for these factors (eg Chen et al, AJHG 2016, PMID:27018471)

In general, the authors should be more careful in the interpretation of the data. Acknowledging the strengths of their study, they should also recognize potential weaknesses. Out of 88 loci, 29 loci show nominally significant heterogeneity between both studies which is well beyond the expected number. Likewise, only 11 loci reach the Bonferroni corrected p-value for association with asthma in the replication study. It is true that the replication study has less power, but this is attributed to the study design. A combined analysis of both data sets might have been the better choice. All these points need to be discussed in the manuscript, not in materials and methods. Non-replicating loci with an opposite effect direction in the replication study could be assessed as potentially new loci. Finally, loci which are strongly associated with overlapping allergic phenotypes such as eczema could have been identified due to this concomitant allergic disease since allergic co-morbidities were not excluded from the study. This should also be included in the discussion.

Minor points

Table 1 still contains known allergy loci ([ADAD1],IL2-[]-IL21) reported by Ferreira et al (Nat Genet. 2017 Dec;49(12):1752-1757).

In Supplementary Table 2, for SNPs in high LD with previously reported SNPs or for identical SNPs, the authors should clearly state whether the same risk allele replicates or not (additional column). If no data are available in the GWAS catalog, please also check the original manuscripts (e.g. Pickrell et al report alleles and effect directions in the supplementary data).

In Supplementary Table 7, the authors should add the p-values and OR of the coding SNPs.

Reviewer #2:

Remarks to the Author:

The authors have satisfactorily responded to all my comments.

Reviewer #3:

Remarks to the Author:

The experimental part improved greatly after revision. The data is sound.

I have just one small comment. It would be better for readers if the statistical evaluation is included in the figure 2 legend. Currently is just mentioned in the text

Reviewers' comments:

Reviewer #1 (Remarks to the Author):

The authors have addressed most of the comments in an accurate manner and the manuscript has improved considerably.

Comment #1

However, there are still some concerns regarding genomic inflation which seems to be a major problem in both studies according to Supplementary Figure 6. The authors should present the results on genomic inflation in more detail in the results section (genomic inflation factors for both studies) and comment on this in the discussion section. What is the inflation factor in UKB and what are potential reasons for genomic inflation in UK Biobank? Have previous studies reported similar inflation in UKB? If population structure/relatedness is a major concern in the Icelandic study, why have the authors not used a method that effectively controls for these factors (eg Chen et al, AJHG 2016, PMID:27018471)

Response:

Regarding supplementary figure 6 then that QQ-plot primarily shows the excess of associations observed in the datasets due to real association signals. The figure also illustrates how much more power is in the UK Biobank dataset than the Icelandic dataset as the excess is much greater for the former. The figure does not indicate any problems in the studies.

The estimated genomic inflation in the two studies (estimated using LD-score regression to reduce the impact of real association signals) is 1.368 for the Icelandic dataset and 1.092 for the UK dataset. For the Icelandic dataset this inflation is due to relatedness of the study individuals. The Icelandic study includes about 360,000 individuals which is almost the whole Icelandic nation, hence every individual has many relatives in the study. This does not indicate any problem with the study and is adjusted for by reducing the test statistic by the inflation factor. For the UK Biobank relatedness is also the most likely cause of the inflation as population structure is adjusted for by including 40 principal components in the analysis. Although not collected as families, 30% of the participants have a relative (third degree or closer) in the UKB dataset and it includes 22,666 sibling pairs (cite: Bycroft C et.al. Nature 11 cot 2018). Large genomic inflation factors have been reported before for GWAS based on UK Biobank data, e.g. $\lambda_g = 1.146$ in a study on hair color (cite: PMID: 29662168) and 1.23 in a study on intraocular pressure (cite: <https://doi.org/10.1038/s41588-018-0126-8>).

Methods that attempt to account for relatedness in the case control setting have other problems. In particular, rare variant associations are not properly handled and the association we do uses phenotype information from ungenotyped individuals by inferring genotype information from genotyped relatives (sometimes called familial imputation). This we can do as a very large fraction of the Icelandic population is genotyped and this method can increase the number of cases used in the study by about 75%. The familiarly imputed genotypes are less informative than imputed genotypes for genotyped individuals and this has to be appropriately accounted for in the association test (cite: <https://doi.org/10.1038/sdata.2015.11>). We have not yet found a method that consistently performs better than the currently applied one. The method we are currently using could potentially be improved to increase power, but it has the benefit of being robust and not produce false positive associations. We note that we are using the bolt method to account for relatedness for quantitative trait associations.

In our previous response we explained how we calculate the effective sample size for each cohort taking into account the genomic adjustment factor (GC), based on LD-score regression for both cohorts (UK_{GC}=1.092) and Iceland_{GC}=1.368) “In total we had 16,247 asthma cases and 346,486 controls from Iceland and 52,942 cases and 355,713 controls from the UK Biobank. Estimating the effective number of cases as $2 \times N_a \times N_c / (N_a + N_c) / GC$, where N_a and N_c are the number of cases and controls, respectively, and GC is the genomic adjustment factor, estimated using LD-score regression (Bulik-Sullivan, B.K., Nat Gen, 2015), we can calculate the effective sample size for Iceland to be $N_{eff}=22,689$ and UK Biobabank to be $N_{eff}=84,396$, $GC=1.368$ in Iceland and $GC=1.092$ in UK Biobank”. This was already added in the Material and Methods section page 15. To further, bring this into the result session we have added a sentence on this on page 3: “The estimated genomic inflation in the two studes (estimated using LD-score regression)⁷ is 1.368 for the Icelandic dataset and 1.092 for the UK dataset, reflecting the relatedness of individuals in Iceland and is also the most likely explanation of genomic inflation in UK Biobank where 30% of the participants have a relative (third degree or closer) in the UKB dataset⁸.”

Supplementary Figure 6. Quantile-quantile plot (QQ-plot) showing chi-square statistics corrected for genomic control and showing variants found in UK and UK cohorts for the GWAS of Asthma in Iceland (red) the UK (blue) and meta-analysis of the two cohorts (black). The red diagonal line represents expected distribution assuming no inflation of the chi-square statistics.

Comment #2

In general, the authors should be more careful in the interpretation of the data. Acknowledging the strengths of their study, they should also recognize potential weaknesses. Out of 88 loci, 29 loci show nominally significant heterogeneity between both studies which is well beyond the expected number. Likewise, only 11 loci reach the Bonferroni corrected p-value for association with asthma in the replication study. It is true that the replication study has less power, but this is attributed to the study design. A combined analysis of both data sets might have been the better choice. All these points need to be discussed in the manuscript, not in materials and methods. Non-replicating loci with an opposite effect direction in the replication study could be assessed as potentially new loci. Finally, loci which are strongly associated with overlapping allergic phenotypes such as eczema could have been identified due to this concomitant allergic disease since allergic co-morbidities were not excluded from the study. This should also be included in the discussion.

Response:

As clearly stated in the abstract, main text and material and methods, this is a meta-analysis of the two sample sets and therefore, we have indeed performed a combined analysis of both datasets as the reviewer suggested.

Regarding the observed heterogeneity between the effect estimates in the two studies, it is worth noting that these are retrospective studies and the diagnostic criteria, ascertainment and age distribution differs between them. Furthermore, it is likely that environmental exposure differs between the two populations. For these reasons we do not expect the effect estimates to be exactly the same in the two populations and the question if we can detect heterogeneity becomes one of power. Consistent with this scenario, all the variants that show nominally significant heterogeneity are previously published signals, i.e. the signals that have the strongest association. And there is a strong correlation between the strength of the association signals and the significance of the heterogeneity. Thus the observed heterogeneity in the effect estimates probably shows that these two studies are large enough to have good power to detect the real difference in the phenotype definition and/or ascertainment between the studies. This is likely to be true for most meta-analysis of GWAS studies that are being done, where the GWAS studies usually differ in terms of ascertainment and/or phenotype definition; however in most cases the individual studies are not as large as here and the power to detect heterogeneity is much less.

This has been addressed by the following sentence on page 3: „This observed heterogeneity in the effect estimates indicates that these two studies are powered to detect the real difference in the phenotype definition and/or ascertainment between the studies that are coherent with retrospective studies as reported here“.

As pointed out in the result section, 85 out of the 88 variants (which is much more than would be expected randomly, $P < 1 \times 10^{-16}$) have effect size in the same direction in the two sample sets. The three variants that seem to have opposite effect direction are completely flat in Iceland as can be seen in supplementary table 1: rs1684466 (OR=1.00652, P=6.86E-01 in Iceland and OR=0.94365, P=4.79E-15 in UKB), rs17454584 (OR=0.999, P=9.56E-01 in Iceland and OR=1.06375, P=8.15E-14 in UKB), rs41284471 (OR=0.9905, P=6.34E-01 in Iceland and OR=1.05369, P=9.98E-10 in UKB).

The reviewer is correct that we did not exclude allergic co-morbidities from our asthma association studies. However, we note that only 980 individuals of our 16,247 (6%) asthma cases in Iceland have the atopic dermatitis (AD) diagnosis and only approx 100 individuals had the ICD10 code for AD (L20) in the UKB at the time of analysis. However, as this is a retrospective study, not designed to address

co-morbidities of asthma, we cannot exclude that co-morbidities are underreported/diagnosed and that those might affect associations of certain sequence variants with asthma.

This has been addressed by the following paragraph page 9: “Since this is a retrospective study, not designed to study co-morbidities of asthma and allergic diseases, we cannot exclude that some of the overlapping association of certain sequence variants with e.g. asthma and AD might be due to overlap of the two phenotypes. However, only 980 out 16,247 asthma cases in Iceland had the AD dermatitis diagnosis (6%) while AD was hardly reported as a single disease in UKB, making it impossible to accurately define asthmatics without AD.”

Minor points

Comment #3

Table 1 still contains known allergy loci ([ADAD1],IL2-[]-IL21) reported by Ferreira et al (Nat Genet. 2017 Dec;49(12):1752-1757).

Response: The reviewer is right and we apologize for this oversight. We have now moved the ADAD1 locus from Table 1 to supplementary Table 5 where we have summarized other loci reported by Ferreira et al, 2017. We have further, updated the number of novel variants accordingly to 19 at 16 loci and 22 variants at 16 loci that have previously been reported for a combined allergic phenotype

Comment #4

In Supplementary Table 2, for SNPs in high LD with previously reported SNPs or for identical SNPs, the authors should clearly state whether the same risk allele replicates or not (additional column). If no data are available in the GWAS catalog, please also check the original manuscripts (e.g. Pickrell et al report alleles and effect directions in the supplementary data).

Response: Unfortunately, it is not straight forward to understand what the risk allele is from the supplements in Pickrell et al. In the supplementary table for asthma they report the alleles as C/T, G/T etc. which one could believe would be the reference/non-reference allele. That would be in line with their description in the supplementary methods that „Throughout the paper, we report effect sizes of variants as the effect of the non-reference allele in human genome reference hg19.“ However, when we look the SNPs up in ncbi database using GRCh37 (hg19) there is inconsistency for many of the variants with the table reported in Pickrell et al. with regards to which allele is the reference allele. When we blindly follow the instructions that they report the effect of the non-reference allele we do not replicate the allele for any of the variants that are inconsistent with what is reported in the ncbi database. Whereas we replicate the effect for all variants that are in line with the ncbi database e.g. rs1723018 that according to ncbi is an A>G variant and is written as A/G in Pickrell have correlated risk alleles in our data and Pickrell. If we believe that the authors have intended their alleles to be written as reference/non-reference allele and they therefore are always reporting the effect size on the latter allele (whether or not that is the non-reference allele) then we replicate their findings 100% (see Table 1 below). However, it is impossible to tell from the Pickrell paper as it is and the same applies for Wan Yi. 2012 (PMID: 22561531), Torgerson (PMID:21804549) and Almoguera (PMID:27611488) which is probably the reason why the risk allele gets a question

mark in the GWAS catalog. Therefore, we have to stick to what is reported in the GWAS catalog and leave the risk allele with a question mark.

We have added a column for the correlated alleles between SNPs and whether or not we replicate the risk allele for other publications in Supplementary Table 2.

Table 1. Previously reported SNPs from Pickrell_JK, 2016 (PMID:27182965) and correlation of risk alleles with our study.

SNP reported in Pickrell et al	alleles as reported in Pickrell et al	RefPalt alleles as reported in GRCh37*	OR reported in Pickrell	Risk allele in Pickrell**	Risk allele in our data	Correlated with our allele	Correlated alleles***
rs6683383	A/T	T>A	1.062	A	T	No	T=T; A=G
rs1723018	A/G	A>G	0.945	A	G	Yes	A=G; G=A
rs13412757	A/G	G>A	1.062	A	G	No	G=G; A=A
rs34290285	A/G	G>A	1.107	A	G	No	same variant
rs5743618	A/C	C>A	1.082	A	C	No	same variant
rs2244012	A/G	A>G	1.102	G	T	Yes	A=C; G=T
rs10957978	G/T	G>T	0.934	G	A	Yes	G=A; T=G
rs144829310	G/T	G>T	1.165	T	A	Yes	G=G; T=A
rs12413578	C/T	C>T	0.891	C	C	Yes	same variant
rs7936323	A/G	G>A	0.922	G	T	No	G=G; A=T
rs3784099	A/G	G>A	0.942	G	A	No	G=T; A=A
rs10519068	A/G	G>A	1.10	A	C	No	G=C; A=T
rs56375023	A/G	G>A	0.897	G	A	No	G=C; A=A
rs7203459	C/T	T>C	1.092	C	T	No	T=T; C=A
rs11655198	C/T	C>T	0.847	C	G	Yes	C=G; T=T

*non-reference allele according to GRCh37

** Risk alleles in Pickrell defined from their description in supplementary methodology that "Throughout the paper, we report effect sizes of variants as the effect of the non-reference allele in human genome reference hg19."

*** Correlated alleles according to https://idlink.ncl.nih.gov/?var=rs1723018&pop=CEU&r2_d=r2&tab=idproxy

Supplementary Table from Pickrell_JK, 2016 (PMID:27182965).

Index SNPs for Strongest Associations

cytoband	assay.name	scaffold	position	alleles	src	pvalue	OR	95% CI	gene.context
17q12	rs11655198	chr17	38026169	C/T	I	1.0×10^{-63}	0.847	[0.831,0.863]	[ZBPB2]
6p21.32	rs3104367	chr6	32603487	C/T	I	1.0×10^{-40}	0.872	[0.855,0.890]	HLA-DRB5---[]-HLA-DQA1
9p24.1	rs144829310	chr9	6208030	G/T	I	1.3×10^{-31}	1.165	[1.136,1.195]	RANBP6---[]-IL33
5q22.1	rs1837253	chr5	110401872	C/T	I	3.3×10^{-31}	0.876	[0.856,0.896]	SLC25A46---[]-TSLP
2q12.1	rs202011557	chr2	102913642	D/I	I	5.1×10^{-31}	0.842	[0.818,0.868]	IL1RL2--[]--IL1RL1
15q22.23	rs56375023	chr15	67448363	A/G	I	2.4×10^{-21}	0.897	[0.877,0.917]	[SMAD3]
11q13.5	rs7936323	chr11	76293758	A/G	I	1.4×10^{-16}	0.922	[0.905,0.940]	C11orf30--[]--LRRC32
6p21.33	rs2428494	chr6	31322197	A/T	I	1.4×10^{-16}	0.920	[0.902,0.939]	HLA-C--[]--MICA
5q31.1	rs2244012	chr5	131901225	A/G	I	2.1×10^{-16}	1.102	[1.077,1.128]	[RAD50]
2q37.3	rs34290285	chr2	242698640	A/G	I	1.8×10^{-15}	1.107	[1.079,1.135]	[D2HGDH]
16p13.13	rs7203459	chr16	11230703	C/T	I	3.5×10^{-15}	1.092	[1.068,1.117]	[CLEC16A]
1q23.3	rs4233366	chr1	161159147	C/T	I	4.8×10^{-15}	1.090	[1.067,1.114]	B4GALT3--[]ADAMTS4
10p14	rs12413578	chr10	9049253	C/T	I	8.1×10^{-12}	0.891	[0.862,0.921]	GATA3---[]
8q21.13	rs10957978	chr8	81285139	G/T	I	1.1×10^{-11}	0.934	[0.915,0.952]	TPD52---[]--ZBTB10
15q22.2	rs10519068	chr15	61068704	A/G	I	3.8×10^{-11}	1.100	[1.069,1.132]	RORA---[]
4p14	rs5743618	chr4	38798648	A/C	I	3.9×10^{-11}	1.082	[1.057,1.107]	[TLR1]
6q15	rs58521088	chr6	90985198	A/T	I	7.1×10^{-11}	0.934	[0.915,0.954]	[BACH2]
12q13.3	rs3001426	chr12	57509055	C/T	I	1.4×10^{-10}	0.938	[0.919,0.956]	STAT6-[]---LRP1
3q28	rs73196739	chr3	188402471	C/T	I	6.5×10^{-9}	0.922	[0.897,0.948]	[LPP]
1q32.1	rs6683383	chr1	203100504	A/T	I	1.1×10^{-8}	1.062	[1.040,1.084]	[ADORA1]
2p25.1	rs13412757	chr2	8458080	A/G	I	1.3×10^{-8}	1.062	[1.040,1.084]	[]---ID2
1q24.2	rs1723018	chr1	167433420	A/G	I	1.4×10^{-8}	0.945	[0.926,0.963]	[CD247]
14q24.1	rs3784099	chr14	68749927	A/G	I	1.6×10^{-8}	0.942	[0.922,0.961]	[RAD51B]
7q22.3	rs6959584	chr7	105676505	C/T	I	2.0×10^{-8}	1.086	[1.055,1.117]	[CDHR3]
5q31.3	rs200634877	chr5	141529761	D/I	I	2.5×10^{-8}	0.940	[0.919,0.961]	[NDFIP1]
1q25.1	rs6691738	chr1	173152036	G/T	I	2.9×10^{-8}	0.943	[0.923,0.963]	TNFSF18---[]TNFSF4
1p36.22	rs662064	chr1	10557251	C/T	I	3.2×10^{-8}	0.942	[0.922,0.962]	[PEX14]

Comment #5

In Supplementary Table 7, the authors should add the p-values and OR of the coding SNPs.

Response: This has been added with the addition of EA/OA for the coding SNPs and correlation of alleles between the strongest asthma associated variant and the coding variant.

Reviewer #2 (Remarks to the Author):

The authors have satisfactorily responded to all my comments.

Reviewer #3 (Remarks to the Author):

The experimental part improved greatly after revision. The data is sound.

I have just one small comment. It would be better for readers if the statistical evaluation is included in the figure 2 legend. Currently is just mentioned in the text

Response: This has been added.

Reviewers' comments:

Reviewer #1 (Remarks to the Author):

The authors have addressed most of the comments in an accurate manner and the manuscript has improved considerably.

Comment #1

However, there are still some concerns regarding genomic inflation which seems to be a major problem in both studies according to Supplementary Figure 6. The authors should present the results on genomic inflation in more detail in the results section (genomic inflation factors for both studies) and comment on this in the discussion section. What is the inflation factor in UKB and what are potential reasons for genomic inflation in UK Biobank? Have previous studies reported similar inflation in UKB? If population structure/relatedness is a major concern in the Icelandic study, why have the authors not used a method that effectively controls for these factors (eg Chen et al, AJHG 2016, PMID:27018471)

Response:

Regarding supplementary figure 6 then that QQ-plot primarily shows the excess of associations observed in the datasets due to real association signals. The figure also illustrates how much more power is in the UK Biobank dataset than the Icelandic dataset as the excess is much greater for the former. The figure does not indicate any problems in the studies.

The QQ plot is not informative, because it shows the GC-corrected p-values. Hence, it lacks essential information on genomic inflation. In order to assess the characteristics of a p-value distribution and thus the validity of potentially true association signals, the authors should present the uncorrected p-values in Supplementary Figure 6.

The estimated genomic inflation in the two studies (estimated using LD-score regression to reduce the impact of real association signals) is 1.368 for the Icelandic dataset and 1.092 for the UK dataset. For the Icelandic dataset this inflation is due to relatedness of the study individuals. The Icelandic study includes about 360,000 individuals which is almost the whole Icelandic nation, hence every individual has many relatives in the study. This does not indicate any problem with the study and is adjusted for by reducing the test statistic by the inflation factor.

Indeed, there seems to be high degree of relatedness/population structure present in the Icelandic data set which accounts for genomic inflation. A high LD-score regression intercept indicates that genomic inflation is not due to true association signals but mainly due to population stratification. This is in contradiction to the first paragraph of the response where the authors state that „the excess of associations observed in the datasets“ is „due to real association signals.“

Genomic inflation needs to be discussed in more detail because it might affect the validity of the association results which is particularly true for the Icelandic data set. This could be a minor issue for common variants but may affect the false positive rate for low frequency and rare variants.

For the UK Biobank relatedness is also the most likely cause of the inflation as population structure is adjusted for by including 40 principal components in the analysis. Although not collected as families, 30% of the participants have a relative (third degree or closer) in the UKB dataset and it includes 22,666 sibling pairs (cite: Bycroft C et.al. Nature 11 Oct 2018). Large genomic inflation factors have

been reported before for GWAS based on UK Biobank data, e.g. $\lambda_g = 1.146$ in a study on hair color (cite: PMID: 29662168) and 1.23 in a study on intraocular pressure (cite: <https://doi.org/10.1038/s41588-018-0126-8>).

Both UKBB studies referenced here excluded related individuals. The IOP study reported a high lambda but a lower LD-score regression intercept. Likewise, the recently published UKBB GWAS on asthma (Johansson et al. 2019) excluded related individuals yielding an improved LD-score regression intercept of 1.065 for the same data set. Excluding related individuals is a standard QC step if association testing does not account for relatedness. Nonetheless, since the results presented by the authors mainly rely on slightly inflated UKBB data, the analysis should be accepted as it is.

Methods that attempt to account for relatedness in the case control setting have other problems. In particular, rare variant associations are not properly handled and the association we do uses phenotype information from ungenotyped individuals by inferring genotype information from genotyped relatives (sometimes called familial imputation). This we can do as a very large fraction of the Icelandic population is genotyped and this method can increase the number of cases used in the study by about 75%. The familiarly imputed genotypes are less informative than imputed genotypes for genotyped individuals and this has to be appropriately accounted for in the association test (cite: <https://doi.org/10.1038/sdata.2015.11>). We have not yet found a method that consistently performs better than the currently applied one. The method we are currently using could potentially be improved to increase power, but it has the benefit of being robust and not produce false positive associations. We note that we are using the bolt method to account for relatedness for quantitative trait associations.

The use of BOLT is not described in the methods section or in the paper. For which analysis have the authors used BOLT LMM which is only listed in the "Code availability" section? If the familiarly imputed genotypes are used in LMM methods for quantitative traits to account for relatedness why not on qualitative traits? Why is relatedness a problem for quantitative but not for qualitative traits?

Given that the UKBB carries most of the associations on its own, the Icelandic cohort has a lot less weight, and the issue of genomic inflation in the Icelandic set could be left to a proper discussion of the issue. The authors should explain the advantages of their method including of not accounting for relatedness (above paragraph) in detail in the manuscript (discussion section). A discussion on advantages and disadvantages is required (including references).

In our previous response we explained how we calculate the effective sample size for each cohort taking into account the genomic adjustment factor (GC), based on LD-score regression for both cohorts ($UK_{GC}=1.092$) and $Iceland_{GC}=1.368$) "In total we had 16,247 asthma cases and 346,486 controls from Iceland and 52,942 cases and 355,713 controls from the UK Biobank. Estimating the effective number of cases as $2 \times N_a \times N_c / (N_a + N_c) / GC$, where N_a and N_c are the number of cases and controls, respectively, and GC is the genomic adjustment factor, estimated using LD-score regression (Bulik-Sullivan, B.K., Nat Gen, 2015), we can calculate the effective sample size for Iceland to be $N_{eff}=22,689$ and UK Biobank to be $N_{eff}=84,396$, $GC=1.368$ in Iceland and $GC=1.092$ in UK Biobank". This was already added in the Material and Methods section page 15. To further, bring this into the result section we have added a sentence on this on page 3: " The estimated genomic inflation in the two studies (estimated using LD-score regression)⁷ is 1.368 for the Icelandic dataset and 1.092 for the UK dataset, reflecting the relatedness of individuals in Iceland and is also the most likely explanation of genomic inflation in UK Biobank where 30% of the participants have a relative (third degree or closer) in the UKB dataset⁸."

Supplementary Figure 6. Quantile-quantile plot (QQ-plot) showing chi-square statistics corrected for genomic control and showing variants found in UK and UK cohorts for the GWAS of Asthma in Iceland (red) the UK (blue) and meta-analysis of the two cohorts (black). The red diagonal line represents expected distribution assuming no inflation of the chi-square statistics.

Again, the authors should use uncorrected p-values for Supplementary Figure 6.

Comment #2

In general, the authors should be more careful in the interpretation of the data. Acknowledging the strengths of their study, they should also recognize potential weaknesses. Out of 88 loci, 29 loci show nominally significant heterogeneity between both studies which is well beyond the expected number. Likewise, only 11 loci reach the Bonferroni corrected p-value for association with asthma in the replication study. It is true that the replication study has less power, but this is attributed to the study design. A combined analysis of both data sets might have been the better choice. All these points need to be discussed in the manuscript, not in materials and methods. Non-replicating loci with an opposite effect direction in the replication study could be assessed as potentially new loci. Finally, loci which are strongly associated with overlapping allergic phenotypes such as eczema could have been identified due to this concomitant allergic disease since allergic co-morbidities were not excluded from the study. This should also be included in the discussion.

Response:

As clearly stated in the abstract, main text and material and methods, this is a meta-analysis of the two sample sets and therefore, we have indeed performed a combined analysis of both datasets as the reviewer suggested.

Regarding the observed heterogeneity between the effect estimates in the two studies, it is worth noting that these are retrospective studies and the diagnostic criteria, ascertainment and age distribution differs between them. Furthermore, it is likely that environmental exposure differs between the two populations. For these reasons we do not expect the effect estimates to be exactly the same in the two populations and the question if we can detect heterogeneity becomes one of power. Consistent with this scenario, all the variants that show nominally significant heterogeneity are previously published signals, i.e. the signals that have the strongest association. And there is a strong correlation between the strength of the association signals and the significance of the heterogeneity. Thus the observed heterogeneity in the effect estimates probably shows that these two studies are large enough to have good power to detect the real difference in the phenotype definition and/or ascertainment between the studies. This is likely to be true for most meta-analysis of GWAS studies that are being done, where the GWAS studies usually differ in terms of ascertainment and/or phenotype definition; however in most cases the individual studies are not as large as here and the power to detect heterogeneity is much less.

This has been addressed by the following sentence on page 3: „This observed heterogeneity in the effect estimates indicates that these two studies are powered to detect the real difference in the phenotype definition and/or ascertainment between the studies that are coherent with retrospective studies as reported here“.

As pointed out in the result section, 85 out of the 88 variants (which is much more than would be expected randomly, $P < 1 \times 10^{-16}$) have effect size in the same direction in the two sample sets. The three variants that seem to have opposite effect direction are completely flat in Iceland as can be seen in supplementary table 1: rs1684466 (OR=1.00652, $P=6.86E-01$ in Iceland and OR=0.94365, $P=4.79E-15$ in UKB), rs17454584 (OR=0.999, $P=9.56E-01$ in Iceland and OR=1.06375, $P=8.15E-14$ in UKB), rs41284471 (OR=0.9905, $P=6.34E-01$ in Iceland and OR=1.05369, $P=9.98E-10$ in UKB).

The reviewer is correct that we did not exclude allergic co-morbidities from our asthma association studies. However, we note that only 980 individuals of our 16,247 (6%) asthma cases in Iceland have the atopic dermatitis (AD) diagnosis and only approx 100 individuals had the ICD10 code for AD (L20) in the UKB at the time of analysis. However, as this is a retrospective study, not designed to address

co-morbidities of asthma, we cannot exclude that co-morbidities are underreported/diagnosed and that those might affect associations of certain sequence variants with asthma.

This has been addressed by the following paragraph page 9: “Since this is a retrospective study, not designed to study co-morbidities of asthma and allergic diseases, we cannot exclude that some of the overlapping association of certain sequence variants with e.g. asthma and AD might be due to overlap of the two phenotypes. However, only 980 out 16,247 asthma cases in Iceland had the AD dermatitis diagnosis (6%) while AD was hardly reported as a single disease in UKB, making it impossible to accurately define asthmatics without AD.”

Minor points

Comment #3

Table 1 still contains known allergy loci ([ADAD1],IL2-[]-IL21) reported by Ferreira et al (Nat Genet. 2017 Dec;49(12):1752-1757).

Response: The reviewer is right and we apologize for this oversight. We have now moved the ADAD1 locus from Table 1 to supplementary Table 5 where we have summarized other loci reported by Ferreira et al, 2017. We have further, updated the number of novel variants accordingly to 19 at 16 loci and 22 variants at 16 loci that have previously been reported for a combined allergic phenotype

Comment #4

In Supplementary Table 2, for SNPs in high LD with previously reported SNPs or for identical SNPs, the authors should clearly state whether the same risk allele replicates or not (additional column). If no data are available in the GWAS catalog, please also check the original manuscripts (e.g. Pickrell et al report alleles and effect directions in the supplementary data).

Response: Unfortunately, it is not straight forward to understand what the risk allele is from the supplements in Pickrell et al. In the supplementary table for asthma they report the alleles as C/T, G/T etc. which one could believe would be the reference/non-reference allele. That would be in line with their description in the supplementary methods that „Throughout the paper, we report effect sizes of variants as the effect of the non-reference allele in human genome reference hg19.“ However, when we look the SNPs up in ncbi database using GRCh37 (hg19) there is inconsistency for many of the variants with the table reported in Pickrell et al. with regards to which allele is the reference allele. When we blindly follow the instructions that they report the effect of the non-reference allele we do not replicate the allele for any of the variants that are inconsistent with what is reported in the ncbi database. Whereas we replicate the effect for all variants that are in line with the ncbi database e.g. rs1723018 that according to ncbi is an A>G variant and is written as A/G in Pickrell have correlated risk alleles in our data and Pickrell. If we believe that the authors have intended their alleles to be written as reference/non-reference allele and they therefore are always reporting the effect size on the latter allele (whether or not that is the non-reference allele) then we replicate their findings 100% (see Table 1 below). However, it is impossible to tell from the Pickrell paper as it is and the same applies for Wan YI. 2012 (PMID: 22561531), Torgerson (PMID:21804549) and Almoguera (PMID:27611488) which is probably the reason why the risk allele gets a question

mark in the GWAS catalog. Therefore, we have to stick to what is reported in the GWAS catalog and leave the risk allele with a question mark.

We have added a column for the correlated alleles between SNPs and whether or not we replicate the risk allele for other publications in Supplementary Table 2.

Table 1. Previously reported SNPs from Pickrell_JK, 2016 (PMID:27182965) and correlation of risk alleles with our study.

SNP reported in Pickrell et al	alleles as reported in Pickrell et al	Ref-bait alleles as reported in GRCh37*	OR reported in Pickrell	Risk allele in Pickrell**	Risk allele in our data	Correlated with our allele	Correlated alleles**
rs6683383	A/T	T>A	1.062	A	T	No	T=T, A=G
rs1723018	A/G	A>G	0.945	A	G	Yes	A=G, G=A
rs13412757	A/G	G>A	1.062	A	G	No	G=G, A=A
rs34290285	A/G	G>A	1.107	A	G	No	same variant
rs5743618	A/C	C>A	1.082	A	C	No	same variant
rs2244012	A/G	A>G	1.102	G	T	Yes	A=C, G=T
rs10957978	G/T	G>T	0.934	G	A	Yes	G=A, T=G
rs144829310	G/T	G>T	1.165	T	A	Yes	G=G, T=A
rs12413578	C/T	C>T	0.891	C	C	Yes	same variant
rs7936323	A/G	G>A	0.922	G	T	No	G=G, A=T
rs3784099	A/G	G>A	0.942	G	A	No	G=T, A=A
rs10519068	A/G	G>A	1.10	A	C	No	G=C, A=T
rs56375023	A/G	G>A	0.897	G	A	No	G=C, A=A
rs7203459	C/T	T>C	1.092	C	T	No	T=T, C=A
rs11655198	C/T	C>T	0.847	C	G	Yes	C=G, T=T

*non-reference allele according to GRCh37
** Risk alleles in Pickrell defined from their description in supplementary methodology that "Throughout the paper we report effect sizes of variants as the effect of the non-reference allele in human genome reference hg19."
*** Correlated alleles according to <https://idlink.nih.gov/?var=rs1723018&pop=CEU&r2=d=r2&tab=ldproxy>

Supplementary Table from Pickrell_JK, 2016 (PMID:27182965).

Index SNPs for Strongest Associations

cytoband	assay.name	scaffold	position	alleles	src	pvalue	OR	95% CI	gene.context
17q12	rs11655198	chr17	38026169	C/T	1	1.0×10^{-63}	0.847	[0.831,0.863]	[ZPBP2]
6p21.32	rs3104367	chr6	32603487	C/T	1	1.0×10^{-40}	0.872	[0.855,0.890]	HLA-DRB5---[]-HLA-DQA1
9p24.1	rs144829310	chr9	6208030	G/T	1	1.3×10^{-31}	1.165	[1.136,1.195]	RANBP6---[]-IL33
5q22.1	rs1837253	chr5	110401872	C/T	1	3.3×10^{-31}	0.876	[0.856,0.896]	SLC25A46---[]-TSLP
2q12.1	rs202011557	chr2	102913642	D/I	1	5.1×10^{-31}	0.842	[0.818,0.868]	IL1RL2--[]-IL1RL1
15q22.33	rs56375023	chr15	67448363	A/G	1	2.4×10^{-21}	0.897	[0.877,0.917]	[SMAD3]
11q13.5	rs7936323	chr11	76293758	A/G	1	1.4×10^{-16}	0.922	[0.905,0.940]	C11orf30--[]-LRRC32
6p21.33	rs2428494	chr6	31322197	A/T	1	1.4×10^{-16}	0.920	[0.902,0.939]	HLA-C--[]-MICA
5q31.1	rs2244012	chr5	131901225	A/G	1	2.1×10^{-16}	1.102	[1.077,1.128]	[RAD50]
2q37.3	rs34290285	chr2	242698640	A/G	1	1.8×10^{-15}	1.107	[1.079,1.135]	[D2HGDH]
16p13.13	rs7203459	chr16	11230703	C/T	1	3.5×10^{-15}	1.092	[1.068,1.117]	[CLEC16A]
1q23.3	rs4233366	chr1	161159147	C/T	1	4.8×10^{-15}	1.090	[1.067,1.114]	B4GALT3--[]ADAMTS4
10p14	rs12413578	chr10	9049253	C/T	1	8.1×10^{-12}	0.891	[0.862,0.921]	GATA3---[]
8q21.13	rs10957978	chr8	81285139	G/T	1	1.1×10^{-11}	0.934	[0.915,0.952]	TPD52---[]--ZBTB10
15q22.2	rs10519068	chr15	61068704	A/G	1	3.8×10^{-11}	1.100	[1.069,1.132]	RORA---[]
4p14	rs5743618	chr4	38798648	A/C	1	3.9×10^{-11}	1.082	[1.057,1.107]	[TLR1]
6q15	rs58521088	chr6	90985198	A/T	1	7.1×10^{-11}	0.934	[0.915,0.954]	[BACH2]
12q13.3	rs3001426	chr12	57509055	C/T	1	1.4×10^{-10}	0.938	[0.919,0.956]	STAT6-[]-LRP1
3q28	rs73196739	chr3	188402471	C/T	1	6.5×10^{-9}	0.922	[0.897,0.948]	[LPP]
1q32.1	rs6683383	chr1	203100504	A/T	1	1.1×10^{-8}	1.062	[1.040,1.084]	[ADORA1]
2p25.1	rs13412757	chr2	8458080	A/G	1	1.3×10^{-8}	1.062	[1.040,1.084]	[]-ID2
1q24.2	rs1723018	chr1	167433420	A/G	1	1.4×10^{-8}	0.945	[0.926,0.963]	[CD247]
14q24.1	rs3784099	chr14	68749927	A/G	1	1.6×10^{-8}	0.942	[0.922,0.961]	[RAD51B]
7q22.3	rs6959584	chr7	105676505	C/T	1	2.0×10^{-8}	1.086	[1.055,1.117]	[CDHR3]
5q31.3	rs200634877	chr5	141529761	D/I	1	2.5×10^{-8}	0.940	[0.919,0.961]	[NDFIP1]
1q25.1	rs6691738	chr1	173152036	G/T	1	2.9×10^{-8}	0.943	[0.923,0.963]	TNFSF18---[]TNFSF4
1p36.22	rs662064	chr1	10557251	C/T	1	3.2×10^{-8}	0.942	[0.922,0.962]	[PEX14]

Comment #5

In Supplementary Table 7, the authors should add the p-values and OR of the coding SNPs.

Response: This has been added with the addition of EA/OA for the coding SNPs and correlation of alleles between the strongest asthma associated variant and the coding variant.

Reviewer #2 (Remarks to the Author):

The authors have satisfactorily responded to all my comments.

Reviewer #3 (Remarks to the Author):

The experimental part improved greatly after revision. The data is sound. I have just one small comment. It would be better for readers if the statistical evaluation is included in the figure 2 legend. Currently is just mentioned in the text

Response: This has been added.

Additional comment:

A UKBB study on asthma has recently been published (Hum Mol Genet. 2019 Jul 30. pii: ddz175. doi: 10.1093/hmg/ddz175) which should be referenced, and novelty of loci adjusted.

Reviewers' comments:

Reviewer #1 (Remarks to the Author):

The authors have addressed most of the comments in an accurate manner and the manuscript has improved considerably.

Comment #1

However, there are still some concerns regarding genomic inflation which seems to be a major problem in both studies according to Supplementary Figure 6. The authors should present the results on genomic inflation in more detail in the results section (genomic inflation factors for both studies) and comment on this in the discussion section. What is the inflation factor in UKB and what are potential reasons for genomic inflation in UK Biobank? Have previous studies reported similar inflation in UKB? If population structure/relatedness is a major concern in the Icelandic study, why have the authors not used a method that effectively controls for these factors (eg Chen et al, AJHG 2016, PMID:27018471)

Response:

Regarding supplementary figure 6 then that QQ-plot primarily shows the excess of associations observed in the datasets due to real association signals. The figure also illustrates how much more power is in the UK Biobank dataset than the Icelandic dataset as the excess is much greater for the former. The figure does not indicate any problems in the studies.

The QQ plot is not informative, because it shows the GC-corrected p-values. Hence, it lacks essential information on genomic inflation. In order to assess the characteristics of a p-value distribution and thus the validity of potentially true association signals, the authors should present the uncorrected p-values in Supplementary Figure 6.

We have added panels to Supplementary Figure 6 with the uncorrected P-values.

The estimated genomic inflation in the two studies (estimated using LD-score regression to reduce the impact of real association signals) is 1.368 for the Icelandic dataset and 1.092 for the UK dataset. For the Icelandic dataset this inflation is due to relatedness of the study individuals. The Icelandic study includes about 360,000 individuals which is almost the whole Icelandic nation, hence every individual has many relatives in the study. This does not indicate any problem with the study and is adjusted for by reducing the test statistic by the inflation factor.

Indeed, there seems to be high degree of relatedness/population structure present in the Icelandic data set which accounts for genomic inflation. A high LD-score regression intercept indicates that genomic inflation is not due to true association signals but mainly due to population stratification. This is in contradiction to the first paragraph of the response where the authors state that „the excess of associations observed in the datasets“ is „due to real association signals.“ Genomic inflation needs to be discussed in more detail because it might affect the validity of the association results which is particularly true for the Icelandic data set. This could be a minor issue for common variants but may affect the false positive rate for low frequency and rare variants.

The reviewer is correct, the high correction factor in the Icelandic dataset is because of relatedness between the individuals. The first paragraph in the response stating that „the excess of associations observed in the datasets“ is „due to real association signals“ was incorrect. We have added QQ plots

for asthma stratified on minor allele frequency to the supplement (Supplementary Figure 6 c-d). These show that there is not a systematic problem with false positive at lower frequencies.

For the UK Biobank relatedness is also the most likely cause of the inflation as population structure is adjusted for by including 40 principle components in the analysis. Although not collected as families, 30% of the participants have a relative (third degree or closer) in the UKB dataset and it includes 22,666 sibling pairs (cite: Bycroft C et.al. Nature 11 cot 2018). Large genomic inflation factors have been reported before for GWAS based on UK Biobank data, e.g $\lambda_g = 1.146$ in a study on hair color (cite: PMID: 29662168) and 1.23 in a study on intraocular pressure (cite: <https://doi.org/10.1038/s41588-018-0126-8>).

Both UKBB studies referenced here excluded related individuals. The IOP study reported a high lambda but a lower LD-score regression intercept. Likewise, the recently published UKBB GWAS on asthma (Johansson et al. 2019) excluded related individuals yielding an improved LD-score regression intercept of 1.065 for the same data set. Excluding related individuals is a standard QC step if association testing does not account for relatedness. Nonetheless, since the results presented by the authors mainly rely on slightly inflated UKBB data, the analysis should be accepted as it is.

As explained later we were missing a sentence on on genomic control with LD score regression in the manuscript. We have added this to the methods pg 21: We used LD score regression¹³ to account for inflation in test statistics due to cryptic relatedness and stratification.

Methods that attempt to account for relatedness in the case control setting have other problems. In particular, rare variant associations are not properly handled and the association we do uses phenotype information from ungenotyped individuals by inferring genotype information from genotyped relatives (sometimes called familial imputation). This we can do as a very large fraction of the Icelandic population is genotyped and this method can increase the number of cases used in the study by about 75%. The familiarly imputed genotypes are less informative than imputed genotypes for genotyped individuals and this has to be appropriately accounted for in the association test (cite: <https://doi.org/10.1038/sdata.2015.11>). We have not yet found a method that consistently performs better than the currently applied one. The method we are currently using could potentially be improved to increase power, but it has the benefit of being robust and not produce false positive associations. We note that we are using the bolt method to account for relatedness for quantitative trait associations.

The use of BOLT is not described in the methods section or in the paper. For which analysis have the authors used BOLT LMM which is only listed in the “Code availability” section?

We use BOLT to perform quantitative trait associations that we used for the association with eosinophil count. We have added a description of this to the methods pg 21.

Association testing and meta-analysis of eosinophil counts

Linear mixed model implemented by BOLT-LMM⁶³ was used to test for association between sequence variants and eosinophil counts, assuming an additive genetic model. We assume that the quantitative measurements follow a normal distribution with a mean that depends linearly on the expected allele at the variant and a variance–covariance matrix proportional to the kinship matrix⁶⁴. We used LD score regression¹³ to account for inflation in test statistics due to cryptic relatedness and stratification. To combine the deCODE and UK Biobank results, we used a fixed-effect inverse variance method based on effect estimates and standard errors⁶⁰. We used a likelihood-ratio test to compute all *P* values.

Further description of the eosinophil count was added pg. 19

We obtained eosinophil counts from three of the largest laboratories in Iceland (measurements performed between the years 1993 and 2015). The circulating eosinophil counts were standardized to a standard normal distribution using quantile-quantile standardization and then adjusted for sex, year of birth and age at measurement, as previously described^{9,10}. A total of 251,307 Icelandic individuals with eosinophil counts were included in the study. In the UKB we had eosinophil counts for a total of 396,822 individuals that were adjusted for sex, year of birth, age at measurement and the first 40 principal components, then inverse normal transformed.

If the familiarly imputed genotypes are used in LMM methods for quantitative traits to account for relatedness why not on qualitative traits? Why is relatedness a problem for quantitative but not for qualitative traits?

Genomic control, either the older median based adjustment or LD score regression, has proven to be a robust method for accounting for relatedness. However, LMM models are potentially more powerful. The “problem” is the same for both types of analysis and primarily leads to loss of power. Our methods for both do not create false positives.

Given that the UKBB carries most of the associations on its own, the Icelandic cohort has a lot less weight, and the issue of genomic inflation in the Icelandic set could be left to a proper discussion of the issue. The authors should explain the advantages of their method including of not accounting for relatedness (above paragraph) in detail in the manuscript (discussion section). A discussion on advantages and disadvantages is required (including references).

We were missing a paragraph on genomic control with LD score regression in manuscript. We have added the following paragraph to the methods pg 20:

We used LD score regression¹³ to account for inflation in test statistics due to cryptic relatedness and stratification. We chose to include related individuals in the analysis as removing related individuals in the heavily genotyped Icelandic population would lead to the removal of most participants. Including related individuals while accounting for genomic control has proven to be a robust method^{10,58,59} that does not create false positives as observed in the QQ plots (supplementary figure 6).

In our previous response we explained how we calculate the effective sample size for each cohort taking into account the genomic adjustment factor (GC), based on LD-score regression for both cohorts (UK_{GC}=1.092) and Iceland_{GC}=1.368) “In total we had 16,247 asthma cases and 346,486 controls from Iceland and 52,942 cases and 355,713 controls from the UK Biobank. Estimating the effective number of cases as $2 \times N_a \times N_c / (N_a + N_c) / GC$, where N_a and N_c are the number of cases and controls, respectively, and GC is the genomic adjustment factor, estimated using LD-score regression (Bulik-Sullivan, B.K., Nat Gen, 2015), we can calculate the effective sample size for Iceland to be $N_{eff}=22,689$ and UK Biobabank to be $N_{eff}=84,396$, $GC=1.368$ in Iceland and $GC=1.092$ in UK Biobank“. This was already added in the Material and Methods section page 15. To further, bring this into the result session we have added a sentence on this on page 3: “ The estimated genomic inflation in the two studes (estimated using LD-score regression)⁷ is 1.368 for the Icelandic dataset and 1.092 for the UK dataset, reflecting the relatedness of individuals in Iceland and is also the most

likely explanation of genomic inflation in UK Biobank where 30% of the participants have a relative (third degree or closer) in the UKB dataset.⁸.”

Supplementary Figure 6. Quantile-quantile plot (QQ-plot) showing chi-square statistics corrected for genomic control and showing variants found in UK and UK cohorts for the GWAS of Asthma in Iceland (red) the UK (blue) and meta-analysis of the two cohorts (black). The red diagonal line represents expected distribution assuming no inflation of the chi-square statistics.

Again, the authors should use uncorrected p-values for Supplementary Figure 6.

Agreed, the following panels have been added to supplementary figure 6.

Comment #2

In general, the authors should be more careful in the interpretation of the data. Acknowledging the strengths of their study, they should also recognize potential weaknesses. Out of 88 loci, 29 loci show nominally significant heterogeneity between both studies which is well beyond the expected number. Likewise, only 11 loci reach the Bonferroni corrected p-value for association with asthma in the replication study. It is true that the replication study has less power, but this is attributed to the study design. A combined analysis of both data sets might have been the better choice. All these points need to be discussed in the manuscript, not in materials and methods. Non-replicating loci with an opposite effect direction in the replication study could be assessed as potentially new loci. Finally, loci which are strongly associated with overlapping allergic phenotypes such as eczema could have been identified due to this concomitant allergic disease since allergic co-morbidities were not excluded from the study. This should also be included in the discussion.

Response:

As clearly stated in the abstract, main text and material and methods, this is a meta-analysis of the two sample sets and therefore, we have indeed performed a combined analysis of both datasets as the reviewer suggested.

Regarding the observed heterogeneity between the effect estimates in the two studies, it is worth noting that these are retrospective studies and the diagnostic criteria, ascertainment and age distribution differs between them. Furthermore, it is likely that environmental exposure differs between the two populations. For these reasons we do not expect the effect estimates to be exactly the same in the two populations and the question if we can detect heterogeneity becomes one of power. Consistent with this scenario, all the variants that show nominally significant heterogeneity are previously published signals, i.e. the signals that have the strongest association. And there is a strong correlation between the strength of the association signals and the significance of the heterogeneity. Thus the observed heterogeneity in the effect estimates probably shows that these two studies are large enough to have good power to detect the real difference in the phenotype definition and/or ascertainment between the studies. This is likely to be true for most meta-analysis of GWAS studies that are being done, where the GWAS studies usually differ in terms of ascertainment and/or phenotype definition; however in most cases the individual studies are not as large as here and the power to detect heterogeneity is much less.

This has been addressed by the following sentence on page 3: „This observed heterogeneity in the effect estimates indicates that these two studies are powered to detect the real difference in the phenotype definition and/or ascertainment between the studies that are coherent with retrospective studies as reported here“.

As pointed out in the result section, 85 out of the 88 variants (which is much more than would be expected randomly, $P < 1 \times 10^{-16}$) have effect size in the same direction in the two sample sets. The three variants that seem to have opposite effect direction are completely flat in Iceland as can be seen in supplementary table 1: rs1684466 (OR=1.00652, P=6.86E-01 in Iceland and OR=0.94365, P=4.79E-15 in UKB), rs17454584 (OR=0.999, P=9.56E-01 in Iceland and OR=1.06375, P=8.15E-14 in UKB), rs41284471 (OR=0.9905, P=6.34E-01 in Iceland and OR=1.05369, P=9.98E-10 in UKB).

The reviewer is correct that we did not exclude allergic co-morbidities from our asthma association studies. However, we note that only 980 individuals of our 16,247 (6%) asthma cases in Iceland have the atopic dermatitis (AD) diagnosis and only approx 100 individuals had the ICD10 code for AD (L20) in the UKB at the time of analysis. However, as this is a retrospective study, not designed to address

co-morbidities of asthma, we cannot exclude that co-morbidities are underreported/diagnosed and that those might affect associations of certain sequence variants with asthma.

This has been addressed by the following paragraph page 9: “Since this is a retrospective study, not designed to study co-morbidities of asthma and allergic diseases, we cannot exclude that some of the overlapping association of certain sequence variants with e.g. asthma and AD might be due to overlap of the two phenotypes. However, only 980 out 16,247 asthma cases in Iceland had the AD dermatitis diagnosis (6%) while AD was hardly reported as a single disease in UKB, making it impossible to accurately define asthmatics without AD.”

Minor points

Comment #3

Table 1 still contains known allergy loci ([ADAD1],IL2-[]-IL21) reported by Ferreira et al (Nat Genet. 2017 Dec;49(12):1752-1757).

Response: The reviewer is right and we apologize for this oversight. We have now moved the ADAD1 locus from Table 1 to supplementary Table 5 where we have summarized other loci reported by Ferreira et al, 2017. We have further, updated the number of novel variants accordingly to 19 at 16 loci and 22 variants at 16 loci that have previously been reported for a combined allergic phenotype

Comment #4

In Supplementary Table 2, for SNPs in high LD with previously reported SNPs or for identical SNPs, the authors should clearly state whether the same risk allele replicates or not (additional column). If no data are available in the GWAS catalog, please also check the original manuscripts (e.g. Pickrell et al report alleles and effect directions in the supplementary data).

Response: Unfortunately, it is not straight forward to understand what the risk allele is from the supplements in Pickrell et al. In the supplementary table for asthma they report the alleles as C/T, G/T etc. which one could believe would be the reference/non-reference allele. That would be in line with their description in the supplementary methods that „Throughout the paper, we report effect sizes of variants as the effect of the non-reference allele in human genome reference hg19.“ However, when we look the SNPs up in ncbi database using GRCh37 (hg19) there is inconsistency for many of the variants with the table reported in Pickrell et al. with regards to which allele is the reference allele. When we blindly follow the instructions that they report the effect of the non-reference allele we do not replicate the allele for any of the variants that are inconsistent with what is reported in the ncbi database. Whereas we replicate the effect for all variants that are in line with the ncbi database e.g. rs1723018 that according to ncbi is an A>G variant and is written as A/G in Pickrell have correlated risk alleles in our data and Pickrell. If we believe that the authors have intended their alleles to be written as reference/non-reference allele and they therefore are always reporting the effect size on the latter allele (whether or not that is the non-reference allele) then we replicate their findings 100% (see Table 1 below). However, it is impossible to tell from the Pickrell paper as it is and the same applies for Wan Yi. 2012 (PMID: 22561531), Torgerson (PMID:21804549) and Almoguera (PMID:27611488) which is probably the reason why the risk allele gets a question

mark in the GWAS catalog. Therefore, we have to stick to what is reported in the GWAS catalog and leave the risk allele with a question mark.

We have added a column for the correlated alleles between SNPs and whether or not we replicate the risk allele for other publications in Supplementary Table 2.

Table 1. Previously reported SNPs from Pickrell_JK, 2016 (PMID:27182965) and correlation of risk alleles with our study.

SNP reported in Pickrell et al	alleles as reported in Pickrell et al	RefPath alleles as reported in GRCh37*	OR reported in Pickrell	Risk allele in Pickrell**	Risk allele in our data	Correlated with our allele	Correlated alleles***
rs6683383	A/T	T>A	1.062	A	T	No	T=T; A=G
rs1723018	A/G	A>G	0.945	A	G	Yes	A=G; G=A
rs13412757	A/G	G>A	1.062	A	G	No	G=G; A=A
rs34290285	A/G	G>A	1.107	A	G	No	same variant
rs5743618	A/C	C>A	1.082	A	C	No	same variant
rs2244012	A/G	A>G	1.102	G	T	Yes	A=C; G=T
rs10957978	G/T	G>T	0.934	G	A	Yes	G=A; T=G
rs144829310	G/T	G>T	1.165	T	A	Yes	G=G; T=A
rs12413578	C/T	C>T	0.891	C	C	Yes	same variant
rs7936323	A/G	G>A	0.922	G	T	No	G=G; A=T
rs3784099	A/G	G>A	0.942	G	A	No	G=T; A=A
rs10519068	A/G	G>A	1.10	A	C	No	G=C; A=T
rs56375023	A/G	G>A	0.897	G	A	No	G=C; A=A
rs7203459	C/T	T>C	1.092	C	T	No	T=T; C=A
rs11655198	C/T	C>T	0.847	C	G	Yes	C=G; T=T

*non-reference allele according to GRCh37
** Risk alleles in Pickrell defined from their description in supplementary methodology that "Throughout the paper, we report effect sizes of variants as the effect of the non-reference allele in human genome reference hg19."
*** Correlated alleles according to https://idlink.nci.nih.gov/?var=rs1723018&pop=CEU&r2_d=2&tab=ldproxy

Supplementary Table from Pickrell_JK, 2016 (PMID:27182965).

Index SNPs for Strongest Associations

cytoband	assay.name	scaffold	position	alleles	src	pvalue	OR	95% CI	gene.context
17q12	rs11655198	chr17	38026169	C/T	I	1.0×10^{-63}	0.847	[0.831,0.863]	[ZPBP2]
6p21.32	rs3104367	chr6	32603487	C/T	I	1.0×10^{-40}	0.872	[0.855,0.890]	HLA-DRB5---[]-HLA-DQA1
9p24.1	rs144829310	chr9	6208030	G/T	I	1.3×10^{-31}	1.165	[1.136,1.195]	RANBP6---[]-IL33
5q22.1	rs1837253	chr5	110401872	C/T	I	3.3×10^{-31}	0.876	[0.856,0.896]	SLC25A46---[]-TSLP
2q12.1	rs202011557	chr2	102913642	D/I	I	5.1×10^{-31}	0.842	[0.818,0.868]	IL1RL2---[]-IL1RL1
15q22.23	rs56375023	chr15	67448363	A/G	I	2.4×10^{-21}	0.897	[0.877,0.917]	[SMAD3]
11q13.5	rs7936323	chr11	76293758	A/G	I	1.4×10^{-16}	0.922	[0.905,0.940]	C11orf30---[]-LRRC32
6p21.33	rs2428494	chr6	31322197	A/T	I	1.4×10^{-16}	0.920	[0.902,0.939]	HLA-C---[]-MICA
5q31.1	rs2244012	chr5	131901225	A/G	I	2.1×10^{-16}	1.102	[1.077,1.128]	[RAD50]
2q37.3	rs34290285	chr2	242698640	A/G	I	1.8×10^{-15}	1.107	[1.079,1.135]	[D2HGDH]
16p13.13	rs7203459	chr16	11230703	C/T	I	3.5×10^{-15}	1.092	[1.068,1.117]	[CLEC16A]
1q23.3	rs4233366	chr1	161159147	C/T	I	4.8×10^{-15}	1.090	[1.067,1.114]	B4GALT3---[]ADAMTS4
10p14	rs12413578	chr10	9049253	C/T	I	8.1×10^{-12}	0.891	[0.862,0.921]	GATA3---[]
8q21.13	rs10957978	chr8	81285139	G/T	I	1.1×10^{-11}	0.934	[0.915,0.952]	TPD52---[]-ZBTB10
15q22.2	rs10519068	chr15	61068704	A/G	I	3.8×10^{-11}	1.100	[1.069,1.132]	RORA---[]
4p14	rs5743618	chr4	38798648	A/C	I	3.9×10^{-11}	1.082	[1.057,1.107]	[TLR1]
6q15	rs58521088	chr6	90985198	A/T	I	7.1×10^{-11}	0.934	[0.915,0.954]	[BACH2]
12q13.3	rs3001426	chr12	57509055	C/T	I	1.4×10^{-10}	0.938	[0.919,0.956]	STAT6-[]-LRP1
3q28	rs73196739	chr3	188402471	C/T	I	6.5×10^{-9}	0.922	[0.897,0.948]	[LPP]
1q32.1	rs6683383	chr1	203100504	A/T	I	1.1×10^{-8}	1.062	[1.040,1.084]	[ADORA1]
2p25.1	rs13412757	chr2	8458080	A/G	I	1.3×10^{-8}	1.062	[1.040,1.084]	[]-ID2
1q24.2	rs1723018	chr1	167433420	A/G	I	1.4×10^{-8}	0.945	[0.926,0.963]	[CD247]
14q24.1	rs3784099	chr14	68749927	A/G	I	1.6×10^{-8}	0.942	[0.922,0.961]	[RAD51B]
7q22.3	rs6959584	chr7	105676505	C/T	I	2.0×10^{-8}	1.086	[1.055,1.117]	[CDHR3]
5q31.3	rs200634877	chr5	141529761	D/I	I	2.5×10^{-8}	0.940	[0.919,0.961]	[NDP1P1]
1q25.1	rs6691738	chr1	173152036	G/T	I	2.9×10^{-8}	0.943	[0.923,0.963]	TNFSF18---[]TNFSF4
1p36.22	rs662064	chr1	10557251	C/T	I	3.2×10^{-8}	0.942	[0.922,0.962]	[PEX14]

Comment #5

In Supplementary Table 7, the authors should add the p-values and OR of the coding SNPs.

Response: This has been added with the addition of EA/OA for the coding SNPs and correlation of alleles between the strongest asthma associated variant and the coding variant.

Reviewer #2 (Remarks to the Author):

The authors have satisfactorily responded to all my comments.

Reviewer #3 (Remarks to the Author):

The experimental part improved greatly after revision. The data is sound. I have just one small comment. It would be better for readers if the statistical evaluation is included in the figure 2 legend. Currently is just mentioned in the text

Response: This has been added.

Additional comment: A UKBB study on asthma has recently been published (Hum Mol Genet. 2019 Jul 30. pii: ddz175. doi: 10.1093/hmg/ddz175) which should be referenced, and novelty of loci adjusted.

Since this paper came out so late in the review process, we have added a sentence referring to this paper and the one included in our 2nd submission that have reported novel asthma loci while our manuscript was in review and label them in Table 1, accordingly, as we already did for the paper of Shrine et al, (2nd submission), which the reviewers accepted.

Pg. 5: We note that during the final review of this manuscript two independent reports published association of asthma with 9 of those 19 variants as indicated in Table 1^{15,16}.